# Synthesis and Biological Evaluation of New Isoxazolyl Steroids as Anti-Prostate Cancer Agents

**DOI:** 10.3390/ijms232113534

**Published:** 2022-11-04

**Authors:** Anton S. Rudovich, Miroslav Peřina, Anastasiya V. Krech, Maria Y. Novozhilova, Anastasia M. Tumilovich, Tatyana V. Shkel, Irina P. Grabovec, Miroslav Kvasnica, Lukáš Mada, Maria G. Zavialova, Arif R. Mekhtiev, Radek Jorda, Vladimir N. Zhabinskii, Vladimir A. Khripach

**Affiliations:** 1Institute of Bioorganic Chemistry, National Academy of Sciences of Belarus, Kuprevich Str., 5/2, 220141 Minsk, Belarus; 2Department of Experimental Biology, Faculty of Science, Palacký University Olomouc, Šlechtitelů 27, 78371 Olomouc, Czech Republic; 3Institute of Biomedical Chemistry, 10 Building 8, Pogodinskaya Str., 119121 Moscow, Russia; 4Institute of Molecular and Translational Medicine, Faculty of Medicine and Dentistry, Palacký University Olomouc, Hněvotínská 5, 77900 Olomouc, Czech Republic

**Keywords:** prostate cancer, androgen signaling, androgen receptor, CYP17A1, isoxazoles, Weinreb amide, LNCaP, LAPC-4, molecular docking

## Abstract

Steroids with a nitrogen-containing heterocycle in the side chain are known as effective inhibitors of androgen signaling and/or testosterone biosynthesis, thus showing beneficial effects for the treatment of prostate cancer. In this work, a series of 3β-hydroxy-5-ene steroids, containing an isoxazole fragment in their side chain, was synthesized. The key steps included the preparation of Weinreb amide, its conversion to acetylenic ketones, and the 1,2- or 1,4-addition of hydroxylamine, depending on the solvent used. The biological activity of the obtained compounds was studied in a number of tests, including their effects on 17α-hydroxylase and 17,20-lyase activity of human CYP17A1 and the ability of selected compounds to affect the downstream androgen receptor signaling. Three derivatives diminished the transcriptional activity of androgen receptor and displayed reasonable antiproliferative activity. The candidate compound, **24j** (17*R*)-17-((3-(2-hydroxypropan-2-yl)isoxazol-5-yl)methyl)-androst-5-en-3β-ol, suppressed the androgen receptor signaling and decreased its protein level in two prostate cancer cell lines, LNCaP and LAPC-4. Interaction of compounds with CYP17A1 and the androgen receptor was confirmed and described by molecular docking.

## 1. Introduction

Prostate cancer (PCa) is the most common cancer in men in developed countries. More than 80 years have passed since Charles Huggins showed that a decrease in androgen levels in patients with PCa causes tumor regression [1]. The androgenic pathway remains the main target of prostate cancer therapies—it plays a major role in the formation and progression of this type of cancer [2]. Therapy has aimed at reducing the content of testosterone in the blood, which can significantly slow down the process of tumor development and alleviate the patient’s condition. Therefore, a number of drugs are used to block the synthesis of androgens in the testes or adrenal cortex as an accepted alternative to surgical intervention (orchiectomy).

The most important step in the biosynthesis of androgens is the conversion of pregnenolone to 17α-OH-pregnenolone, and then to dehydroepiandrosterone (DHEA), secreted by the testes and adrenal cortex [3]. Both reactions proceed with the participation of cytochrome P450 CYP17A1, which combines the functions of 17α-hydroxylase and 17,20-lyase. In 2011, a new CYP17A1 inhibitor, abiraterone, was approved, effective for the treatment of prostate cancer, insensitive to hormone therapy, and reducing the level of testosterone in the blood [4]. Thus, abiraterone (Figure 1), which is a pyridine derivative of DHEA, inhibits two key reactions in the androgen synthesis pathway. The optimal CYP17A1 inhibitor should have significant effect on 17,20-lyase activity, with moderate or no effect towards 17α-hydroxylase activity of the enzyme, to modulate sex steroid biosynthesis with minimal effect on glucocorticoid hormones biosynthesis [5].

During the development of new inhibitors of CYP17A1, a large number of androstane and pregnane derivatives have been introduced containing pyridyl-, picolidine-, pyrazolyl-, imidazolyl-, triazolyl-, isoxazolinyl-, dihydrooxazolinyl-, tetrahydrooxazolinyl-, benzimidazolyl-, and carbamoyl- substituents, mainly in positions 16, 17, and 22 [6,7,8,9,10,11,12,13,14,15,16,17,18]. Galeterone, the most advanced among them and having a multiple mechanism of action, has reached phase III clinical trials [19].

To date, previous studies have shown that steroids with 5-membered rings containing one nitrogen and one oxygen (oxazole or isoxazole) are of great interest in the development of drugs for the treatment of prostate cancer [6,11,12,14,20,21,22,23,24,25]. Thus, isoxazole **2** showed potent and non-competitive inhibition of human microsomal 17β-hydroxylase/C17,20-lyase, with an IC_50_ value of 59 nM, and demonstrated potent and competitive inhibition of 5α-reductase in human prostate microsomes with an IC_50_ value of 33 nM [21]. It was also shown that **1,** at a concentration of 5 μM, exhibits antiandrogenic activity in human prostate cancer cell lines (e.g., LNCaP), preventing the binding of labeled synthetic androgen R1881 (5 nM) to the androgen receptor (AR). Compound **2** had a significant effect on the growth of LNCaP and PC-3 cells, commensurate with that of galeterone [11]. It should be noted that **2** showed no inhibitory potency towards CYP17A1, thus confirming that inhibition of this enzyme is not the only mechanism of anticancer action of such steroids. 

Obviously, further studies of new nitrogen-containing steroids, in particular the investigation of their effect on various signaling pathways involved in the pathological processes of tumor development, are relevant and of great interest. In this regard, the present paper aims (i) to develop synthesis of a series of novel steroidal isoxazoles **3a**,**b** and (ii) to carry out studies of their effects on the 17α-hydroxylase and 17,20-lyase activity of human CYP17A1 and the ability of selected compounds to affect the downstream androgen receptor signaling.

## 2. Results and Discussion

### 2.1. Chemistry

One of the tasks of the present study was to develop an efficient route to regioisomeric isoxazoles **9** and **10** (Figure 1). We envisaged that both isoxazoles could be derived from the same α,β-acetylenic ketones **6** via 1,4- or 1,2-cycloaddition of hydroxylamine followed by the cyclization of intermediates **7** or **8**. The regioselectivity of the addition of hydroxylamine to acetylenic ketones can be controlled by the choice of solvent: in a mixture of tetrahydrofuran-water, the reaction proceeds in a 1,4-manner [26], while in aqueous methanol, 1,2-addition products are formed [27,28]. Ynones in **6** could be made available from the known esters in **5**, which in turn can be prepared from commercial 17-ketones in **4**.

The synthesis of target compounds was initiated with ester **11,** obtained in two steps from androstenolone [29]. Initially, the possibility of a one-step conversion of **12** into **18**, described for fluoroketones [26] and consisting in the addition of lithium acetylenides to esters in the presence of boron trifluoride etherate, was studied (Figure 2). However, the reaction of **12** with a lithium salt of **13** resulted in the formation of a complex mixture of products.

Next, an attempt was made to obtain ynone **18** using an approach based on the conversion of ester **12** to aldehyde **16**. Its reaction with the lithium salt of **13** gave a mixture of isomeric alcohols in **17**, which was oxidized in the last stage to give the target ynone **18**. The obvious disadvantage of this method was the necessity to accomplish a multistep reaction procedure. In this connection, the possibility of using Weinreb amides was studied. This approach showed good results for the preparation of ynone **18** via amide **14** and was further used for the synthesis of all other α,β-acetylenic ketones.

Ynone **20b**–**i** was prepared in a 78–90% yield by the addition of lithium salts obtained in situ from BuLi and the corresponding acetylene **19b**–**i** to the Weinreb amide **14** (Figure 3). The unsubstituted ynone **20a** was synthesized by using commercially available ethynylmagnesium bromide as an organometallic reagent. 

The regioselectivity of hydroxylamine addition to α,β-acetylenic ketones is highly dependent on the reaction conditions. The optimal conditions for the conjugated 1,4-addition were found using a mixture of organic solvents with water [28]. The enamine, formed as a result of conjugated addition, undergoes intramolecular cyclization to form 5-hydroxy-4,5-dihydroisoxazoles. We used the reaction conditions (water–THF, NaHCO_3_ as a base) proposed in [26]; in this case, the hydroxyisoxazoline **22a**–**i** was obtained in a 56–88% yield.

The next stage involved the dehydration of **22a**–**i** to form the corresponding isoxazole **23a**–**i**. The reaction proceeded relatively smoothly in the case of the derivative **22d**–**i**; however, isoxazoles **23b**,**c** were obtained only in 9 and 45% yields, respectively. Simultaneously, compounds **26b**,**c** (44–50%) were isolated from the reaction mixture (Figure 4). The possible mechanism of their formation can be explained as follows. 5-Hydroxy-4,5-dihydroisoxazoles **22b**,**c** are expected to exist in equilibrium with enehydroxylamines **21b**,**c**. The latter, as a result of reaction with CDI, can give derivatives **25b**,**c**. It is known that such derivatives can undergo 3,3-sigmatropic rearrangement [30,31]. In the case of **25b**,**c**, such a rearrangement resulted in the formation of substituted enaminoketones (**26b**,**c)**. Their structures were determined by spectral methods, including two-dimensional NMR experiments. Signals at δ 197.1, 134.2, and 157.2 in the ^13^C NMR spectrum of **26c** were assigned to C-22, C-23, and C-24, respectively. The connectivity of the side chain was established by the key HMBC correlations: H-20 and H-17 correlated to C-22; H-25, H-26, and H-27 correlated to C-24; and H-20 correlated to C-23.

Another direction of the reaction of 5-hydroxy-4,5-dihydroisoxazoles with CDI was found in the case of compound **22a**. In addition to the target isoxazole **23a** (51%), β-oxonitrile **31** was also isolated in 40% yield. A possible mechanism of its formation is shown in Figure 5. It is assumed that the enehydroxylamine **21a** is converted to β-ketoxime **29**, which then reacts with CDI to form the imidazole derivative **30**. The latter loses imidazole carboxylic acid in a six-membered transition state [32] to form β-oxonitrile **31**.

The removal of protective groups completed the synthesis of target isoxazoles **24** containing a steroid at C-5 of the isoxazole heterocycle. The reaction was carried out by treating the esters **23a**–**i** with TBAF or HF. The latter option is preferred for compound **24h**, additionally containing tetrahydropyranyl protection.

Attempts were made to carry out some transformations of the sigmatropic rearrangement product **26c** in order to obtain compounds suitable for biological testing (Figure 4). Removal of the silyl protective group proceeded smoothly, without affecting the functional groups in the side chain, to form alcohol **27**. Removal of the imidazole-carboxylic fragment was expected to be achieved under alkaline hydrolysis conditions. However, the reaction led to compound **28**, containing an oxazolone heterocycle. An attempt to remove the silyl group in **28** (Bu_4_NF/THF) gave a complex mixture of products.

Simultaneous removal of both protecting groups in **24i** gave the diol **32** (Figure 6). Compound **24i** contains a functional group in the isoxazole core that can be used for the synthesis of other derivatives, which was demonstrated in the synthesis of azide **36**. Selective removal of the tetrahydropyranyl protecting group was achieved by reaction of **24i** with magnesium bromide diethyl etherate [33]. The tosylation reaction of **33** gave chloride **37** instead of the expected tosylate. The desired product **35** was obtained via S_N_2 displacement of primary mesylate **34** with the azide group.

Ynone **20d** was used as a model compound to study suitable conditions for the preparation of isoxazoles **40**. Its reaction with hydroxylamine in aqueous methanol in the presence of NaHCO_3_ [27,28] gave oxime **39** as a mixture (1:1) of *E*/*Z*-isomers (Figure 7). The next step in the synthesis of isoxazole **40** was the gold-catalyzed cycloisomerization of acetylenic oximes [34]. The desired product **40d** was obtained from **39**, but in a moderate 46% yield, as only the *Z*-isomer underwent the cyclization under these conditions. At the same time, it was found that prolonged heating of the reaction mixture at the stage of hydroxylamine addition led directly to the formation of isoxazoles without any catalysis. 

For this reason, the transformation of the remaining ynones **20a**,**e**–**i** was carried out in one step without the isolation of the intermediate acetylenic oximes (Figure 8). The yield of isoxazoles **40a**,**e**–**i** was 40–82%, depending on the substituent R. Removal of the silyl protective group was performed out by treatment with Bu_4_NF or, in the case of compounds **40h**,**i**, containing additionally tetrahydropyranyl ether, with HF in a mixture of THF-MeCN.

### 2.2. Biology

Biological studies included analysis of the interaction of the prepared compounds with the CYP17A1 active site, testing their effect towards the AR-transcriptional activity and evaluation of their ability to influence the downstream AR signaling. Compounds used in one or more biological tests are shown in Table 1.

#### 2.2.1. Effect of Compounds on 17α-hydroxylase and 17,20-lyase Activity of Human CYP17A1

As the first step in the analysis of the interaction of compounds with the CYP17A1 active site, we performed spectrophotometric titration of selected compounds. Progesterone with Kd_app_ < 1 μM was used as a positive control for the substrate-like ligand (type I). As a positive control for an inhibitor-like ligand (type II), abiraterone with a Kd_app_ < 1 μM was used. DMSO or ethanol was used as a negative control. For the negative controls, no type I or type II spectral responses were observed. Analysis of binding of series of compounds (**24a**,**d**,**g**,**j**, **32**, **36**, **38**) with human CYP17A1 showed that only four compounds were able to bind to the active site of human CYP17A1 (**32** with Kd_app_-13.41 ± 2.38 µM, **24j** with Kd_app_-1.90 ± 0.23 µM, **24d** with Kd_app_-1.50 ± 0.21 µM, and **24a** with Kd_app_-0.13 ± 0.02 µM). However, all these compounds show type I (substrate-like) spectral response, which indicates their potentially low inhibitory ability against this enzyme.

To evaluate the effect of isoxazoles on a potential molecular target, CYP17A1, we performed an inhibitory assay using an in vitro reconstituted system containing recombinant human CYP17A1. We analyzed the inhibitory effect of the compounds (50 μM—final concentration) on two types of reactions catalyzed by CYP17A1: 17α-hydroxylase activity and 17,20-lyase activity. There was almost no inhibition of 17,20-lyase activity of human CYP17A1 with the compound **24j** (11% of enzyme inhibition at 50 µM of compound **24j**). Most isoxazoles were shown to have a moderate inhibitory effect on human CYP17A1 activity (Table 2). The maximum inhibition of 17,20-lyase activity was found for isoxazole **41a** containing no substituent at C-5 of the heterocyclic ring. A similar inhibitory effect was observed for pyridine derivative **41f**. It should be noted that compound **41a** showed a minimal inhibitory capacity for 17α-hydroxylase activity, being the most 17,20 lyase selective, which is important for the development of the next generation CYP17 targeted drugs [35].

#### 2.2.2. The Effect of Derivatives on AR Transcriptional Activity and Viability of PCa Cells

Based on the structural similarity of novel derivatives with galeterone and other published compounds (abiraterone, **3**, and [6]), we tested the effect of our compounds towards the AR-transcriptional activity. Compounds were evaluated using an AR-dependent reporter cell line (ARE14) with a firefly-luciferase gene under the control of an androgen response element [36].

As shown in Table 3, within the studied isoxazoles, three AR antagonists with moderate activity were found (reduced R1881-stimulated AR transcriptional activity to ≤ 50% at 50 µM concentration), namely **24j**, **32**, and **41a**. Despite that, these derivatives acted as AR-antagonists in dose-dependent manner, and none of them were able to overcome the activity of the standard steroidal antagonist galeterone (≈35% of activity at 10 µM concentration). Based on the obtained results, it is evident that derivatives bearing only unsubstituted isoxazole (**41a**), or isoxazoles substituted with small polar substituent (-CH_3_-OH in **32**, t-butyl in **24j**), were able to decrease the AR-transcriptional activity, while isoxazoles bearing longer unsaturated (**24d**), or bulky aromatic substituents (**24e**, **24g**), were inactive. Importantly, none of novel derivatives displayed AR-agonist activity in the chosen concentrations (Table 3).

Antiproliferative properties of all novel steroids were tested in two AR-positive prostate cancer cell lines (LNCaP and LAPC-4) and one AR-negative cell line (DU145) using the Alamar-blue assay after 72 h treatment. Antiproliferative activities of the most potent derivatives **24j** and **32** displayed mid-micromolar values (in agreement to AR-antagonist assay) in both AR-positive PCa cell lines, while no targeting of the AR-negative DU145 cells was observed. Compound **36** displayed reasonable antiproliferative activity only in LAPC-4 cell line. 

We further evaluated the potency of the most active derivative **24j**. We analyzed its effect on AR transcriptional activity in a broad concentration range and found the IC_50_ value = 21.11 ± 1.07 µM (Figure 2A) while IC_50_ = 7.59 µM for galeterone. On the other hand, galeterone displayed worse antiproliferative activities and effects related to AR signaling. Importantly, no clear agonist activity was observed for **24j** in tested concentrations (Figure 2A).

We also performed colony formation assay (CFA) within 10 days to evaluate the prolonged antiproliferative potency of **24j** in the LAPC-4 cell line. CFA is frequently used for the validation of PCa cell lines growth because of their high doubling time in culture. Our compound, **24j**, significantly inhibited the formation of cell colonies in a dose-dependent manner after 10 days in LAPC-4, already from a 1.56 µM concentration (Figure 2B).

#### 2.2.3. Targeting the AR Signaling Pathway

Further, we evaluated the ability of compounds **24j**, **32**, and **41a** to influence the downstream AR signaling (levels of known transcriptional targets PSA and Nkx3.1) in LAPC-4 and LNCaP cell lines after R1881 stimulation. Western blot analysis (Figure 3) showed that **24j** and **32** were able to markedly suppress R1881 stimulated S81-phosphorylation in both LAPC-4 and LNCaP cell lines after 24 h. Observed effects were accompanied by a profound decrease in Nkx3.1 and PSA protein levels in LAPC-4, while were only limited in LNCaP cells (Figure 3).

Candidate compounds were further tested in the same PCa cell lines (without R1881 activation) for a longer period to monitor the effects on AR stability. Compounds **24j** and **32** (12.5 µM) induced a significant decrease in Nkx3.1 and PSA levels in both LAPC-4 and LNCaP after 48 h. Moreover, they both induced a significant decrease in AR protein level that was comparable to galeterone’s effect.

#### 2.2.4. Molecular Docking into the Active Site of CYP17A1 and into the AR-LBD

The binding of candidate compounds into their cellular targets was evaluated by rigid molecular docking into the crystal structure of human CYP17A1 co-crystalized with abiraterone and heme (PDB:3RUK) and by flexible docking into the AR ligand-binding domain (LBD) from the crystal structure with DHT (PDB:2PIV). 

The best binding pose of **41a** in the active site of CYP17A1 was oriented in nearly the same pose as abiraterone and showed similar binding energy (ΔG_Vina_ = −12.6 kcal/mol and −13.0 kcal/mol, respectively) (Figure 4A,B). The isoxazole ring is oriented towards the Fe^2+^ central ion of heme, similar to the pyridine ring in abiraterone. The most promising compound, **24j**, was modelled into the CYP17A1 as well, with a similar pose as abiraterone and **41a**, but with lower binding energy (ΔG_Vina_ = −10.1 kcal/mol, picture not shown).

In the case of the AR-LBD structure, two key amino acid residues in both extremities of the cavity (Arg752 and Thr877) were set as flexible. The docking of **24j** revealed a pose with extensive binding in AR-LBD, with binding energy comparable to galeterone (ΔG_Vina_ = −10.5 kcal/mol and −10.8 kcal/mol, respectively (Figure 4C,D). The position of the steroid core is conserved. C3-OH on the A-ring forms a typical hydrogen bond with Arg752, while the oxygen atom of the isoxazole ring forms a hydrogen bond with Thr877 at the other extremity of the LBD-cavity. The steroid core is further positioned by hydrophobic interactions with Gln711, Met745, Met746, and Leu 704. The tert-butyl substituent could be hydrophobically binded with Leu701, Phe647, and Leu880, which could be a key for the activity and selectivity of **24j** (Figure 4C).

## 3. Materials and Methods

### 3.1. Chemistry

Commercially available reagents were used without further purification. If necessary, solvents were distilled and dried before use by standard methods. Column chromatography was performed through silica gel (200–300 mesh). Thin layer chromatography (TLC) was performed using Silica gel 60 F_254_ plates and visualized using UV light or phosphomolybdic acid. ^1^H and ^13^C NMR spectra were recorded in CDCl_3_, on a Bruker AVANCE 500 spectrometer. Chemical shifts in ^1^H NMR spectra are reported in parts per million (ppm) on the δ scale from an internal standard of residual non-deuterated solvent in CDCl_3_ (7.26 ppm). Data for ^1^H NMR are reported as follows: chemical shift, multiplicity (s = singlet, d = doublet, t = triplet, q = quartet, m = multiplet, br = broad), coupling constant in Hertz (Hz), and integration. Data for ^13^C NMR spectra are reported in terms of chemical shift in ppm from the central peak of CDCl_3_ (77.16 ppm). High resolution mass spectrometry (HRMS) analysis was performed using a Q Exactive HFX (Thermo Scientific) mass spectrometer in ESI ionization mode.

#### 3.1.1. Methyl 2-(3β-((*tert*-butyldimethylsilyl)oxy)-androst-5-en-17-yl)acetate (**12**)

A solution of alcohol **11** (prepared from androstenolone in two steps according to [29]) (1.92 g, 5.55 mmol), TBSCl (1.25 g, 8.29 mmol), and imidazole (838 mg, 12.3 mmol) in dry DMF (8 mL) was stirred at 90 °C for 12 h. After the reaction was completed, the mixture was diluted with water, the organic layer was separated, and the reaction product was extracted from the aqueous layer with PE. The combined organic extracts were dried with sodium sulfate. Then, the solvent was removed under reduced pressure, and the resulting residue was purified by column chromatography on SiO_2_ (PE/EtOAc, 20:1) to yield ether **12** (2.35 g, 92%) as an oil. ^1^H NMR (500 MHz, CDCl_3_) δ 5.31 (dt, *J* = 5.4, 2.1 Hz, 1H), 3.65 (s, 3H), 3.47 (tt, *J* = 11.0, 4.7 Hz, 1H), 2.37 (dd, *J* = 14.7, 5.1 Hz, 1H), 2.26 (ddd, *J* = 13.5, 10.6, 2.7 Hz, 1H), 2.20–2.13 (m, 1H), 2.12 (dd, *J* = 14.7, 9.8 Hz, 1H), 1.00 (s, 3H), 0.88 (s, 9H), 0.60 (s, 3H), 0.05 (s, 6H). ^13^C NMR (126 MHz, CDCl_3_) δ 174.5, 141.8, 121.2, 72.7, 55.8, 51.6, 50.6, 47.0, 43.0, 42.1, 37.6, 37.4, 36.8, 35.1, 32.2, 32.1, 32.1, 28.3, 26.1 (x3), 24.7, 20.9, 19.6, 18.4, 12.5, −4.4 (x2).

#### 3.1.2. 3β-((*tert*-Butyldimethylsilyl)oxy)-pregn-5-en-21-ol (**15**)

A mixture of ester **12** (95 mg, 0.27 mmol), LiAlH_4_ (20 mg, 0.53 mmol), and THF (2 mL) was stirred at 0 °C for 30 min. Then saturated Na_2_SO_4_ (1 mL) and PE (3 mL) were added, and the precipitate was filtered off and washed with PE. The organic layer was evaporated to give alcohol **15** (89 mg, 99%) as an oil. ^1^H NMR (500 MHz, CDCl_3_) δ 5.31 (dt, *J* = 5.0, 2.0 Hz, 1H), 3.68 (ddd, *J* = 10.4, 8.3, 4.6 Hz, 1H), 3.60 (dt, *J* = 10.3, 6.9 Hz, 1H), 3.48 (tt, *J* = 11.1, 4.8 Hz, 1H), 2.26 (tq, *J* = 11.0, 2.7 Hz, 1H), 2.16 (ddd, *J* = 13.3, 5.0, 2.3 Hz, 1H), 1.98 (dtd, *J* = 17.1, 5.1, 2.7 Hz, 1H), 1.00 (s, 3H), 0.88 (s, 9H), 0.60 (s, 3H), 0.05 (s, 6H). ^13^C NMR (126 MHz, CDCl_3_) δ 141.8, 121.2, 72.8, 62.8, 56.1, 50.8, 47.2, 43.0, 42.1, 37.8, 37.6, 36.8, 33.7, 32.2, 32.2, 32.1, 28.5, 26.1 (x3), 25.0, 21.0, 19.6, 18.4, 12.6, −4.4 (x2).

#### 3.1.3. 3β-((*tert*-Butyldimethylsilyl)oxy)-pregn-5-en-21-al (**16**)

A mixture of alcohol **15** (100 mg, 0.23 mmol), Dess-Martin reagent (850 mg, 2.0 mmol), and DCM (4.6 mL) was stirred under argon at 0 °C for 30 min. Then it was diluted with water and extracted with a mixture of PE/EtOAc (3:1). The residue after evaporation of the solvents was chromatographed on SiO_2_ (PE/EtOAc, 90:1→70:30) to afford aldehyde **16** (64 mg, 65%) as an oil. ^1^H NMR (500 MHz, CDCl_3_) δ 9.76 (t, *J* = 2.4 Hz, 1H), 5.31 (dd, *J* = 5.0, 2.5 Hz, 1H), 3.47 (tt, *J* = 11.1, 4.7 Hz, 1H), 1.00 (s, 3H), 0.88 (s, 9H), 0.61 (s, 3H), 0.05 (s, 6H).

#### 3.1.4. 5-((*tert*-Butyldimethylsilyl)oxy)-1-((17*R*)-3β-((*tert*-butyldimethylsilyl)oxy)-androst-5-en-17-yl)pent-3-yn-2-one (**18**)

A 2M solution of BuLi in hexanes (0.7 mL, 1.4 mmol) was added to a cooled −70 °C solution of *tert*-butyldimethyl(prop-2-yn-1-yloxy)silane (**13**) (prepared according to [37], 211 mg, 1.24 mmol) in THF (4 mL). After 15 min, a solution of aldehyde **16** (380 mg, 0.88 mmol) in THF (2.5 mL) was added to the reaction mixture. It was stirred for 20 min at −70 °C, then the cooling bath was removed, and the mixture was allowed to get ambient temperature. NH_4_Cl (150 mg) was added, then the mixture was diluted with water and extracted with EtOAc. The organic layers were dried over Na_2_SO_4_, then evaporated, and the residue was chromatographed on SiO_2_ to give a mixture of isomeric at C-22 alcohol **17** (467 mg), which was used in the next step without further purification.

A mixture of alcohol **17** (467 mg, 0.78 mmol), Dess-Martin reagent (2.30 g, 5.42 mmol), and DCM (15 mL) was stirred under argon at 0 °C for 3 h. Then it was diluted with water and extracted with DCM. The residue after evaporation of the extracts was chromatographed on SiO_2_ (PE/EtOAc, 98:2) to give ketone **18** (388 mg, 73% from **16**) as an oil. ^1^H NMR (500 MHz, CDCl_3_) δ 5.32 (dt, *J* = 5.3, 2.1 Hz, 1H), 4.47 (s, 1H), 3.48 (tt, *J* = 11.1, 4.7 Hz, 1H), 2.63 (dd, *J* = 15.2, 4.0 Hz, 1H), 2.43–2.33 (m, 1H), 2.26 (ddd, *J* = 13.8, 10.8, 2.8 Hz, 1H), 2.17 (ddd, *J* = 13.3, 5.1, 2.3 Hz, 1H), 2.04–1.87 (m, 1H), 1.81 (dt, *J* = 13.3, 3.6 Hz, 1H), 1.00 (s, 3H), 0.92 (s, 9H), 0.89 (s, 9H), 0.61 (s, 3H), 0.14 (s, 6H), 0.05 (s, 6H). ^13^C NMR (126 MHz, CDCl_3_) δ 188.1, 141.8, 121.1, 90.3, 84.3, 72.7, 55.7, 51.7, 50.6, 46.7, 46.2, 43.0, 42.3, 37.6, 37.5, 36.8, 32.2, 32.1, 32.1, 28.2, 26.1 (x3), 25.9 (x3), 24.8, 20.9, 19.6, 18.4, 12.7, −4.4 (x2), −5.0 (x2).

#### 3.1.5. 2-(3β-((*tert*-Butyldimethylsilyl)oxy)-androst-5-en-17-yl)-N-methoxy-N-methylacetamide (**14**)

To a stirred suspension of Weinreb salt (555 mg, 5.72 mmol) in dry toluene (10 mL), a 1M solution of Me_3_Al in heptane (5.7 mL, 5.7 mmol) was added dropwise at 0 °C. After stirring for 40 min at this temperature, a solution of ester **12** (1.00 g, 2.17 mmol) in toluene (10 mL) was added dropwise. The cooling bath was removed, and the reaction mixture was allowed to stir overnight at room temperature. A 2N solution of HCl was added on cooling until pH 2 was reached. the mixture was diluted with water and extracted with EtOAc. The combined organic extracts were washed with saturated aqueous NaHCO_3_, saturated NaCl, dried over anhydrous Na_2_SO_4_, and evaporated to dryness. The residue was purified by silica gel column chromatography (PE/EtOAc, 100:0→70:30) to give the Weinreb amide **14** (650 mg, 61%) as an off-white solid. ^1^H NMR (500 MHz, CDCl_3_) δ 5.31 (dt, *J* = 5.3, 2.0 Hz, 1H), 3.68 (s, 3H), 3.47 (tt, *J* = 10.9, 4.7 Hz, 1H), 3.16 (s, 3H), 2.48 (dd, *J* = 15.1, 4.2 Hz, 1H), 2.25 (qd, *J* = 13.3, 6.4 Hz, 2H), 2.16 (ddd, *J* = 13.3, 5.0, 2.3 Hz, 1H), 1.00 (s, 3H), 0.88 (s, 9H), 0.63 (s, 3H), 0.05 (s, 6H). ^13^C NMR (126 MHz, CDCl_3_) δ 174.9, 141.7, 121.2, 61.3, 55.7, 50.6, 46.4, 43.0, 42.1, 37.6, 37.4, 36.8, 32.6, 32.2, 32.1, 32.1, 28.5, 26.1 (x3), 24.8, 20.9, 19.6, 18.4, 12.7, −4.4 (x2).

#### 3.1.6. 1-(((17*R*)-3β-((*tert*-Butyldimethylsilyl)oxy)-androst-5-en-17-yl)but-3-yn-2-one (**20a**)

To a solution of **14** (300 mg, 0.61 mmol) in THF (0.6 mL), a 0.5M THF solution of ethynylmagnesium bromide (2.8 mL, 1.4 mmol) was added at 0 °C. The reaction mixture was left to warm to room temperature for 1 h, then quenched with saturated NH_4_Cl, and extracted with EtOAc. Combined organics were washed with water, brine, dried over Na_2_SO_4_, and concentrated in vacuo. The crude product was purified by silica gel chromatography (PE:EtOAc, 100:0 to 85:15) to give ynone **20a** (221 mg, 79%) as a white solid. ^1^H NMR (500 MHz, CDCl_3_) δ 5.31 (dt, *J* = 5.1, 2.1 Hz, 1H), 3.48 (tt, *J* = 11.0, 4.8 Hz, 1H), 3.20 (s, 1H), 2.66 (dd, *J* = 15.4, 4.1 Hz, 1H), 2.45–2.33 (m, 1H), 2.26 (ddd, *J* = 13.6, 10.9, 2.7 Hz, 1H), 2.17 (ddd, *J* = 13.3, 5.0, 2.3 Hz, 1H), 2.03–1.86 (m, 2H), 1.81 (dt, *J* = 13.3, 3.6 Hz, 1H), 1.00 (s, 3H), 0.88 (s, 9H), 0.61 (s, 3H), 0.05 (s, 6H). ^13^C NMR (126 MHz, CDCl_3_) δ 187.9, 141.8, 121.1, 81.9, 78.4, 72.7, 55.7, 50.6, 46.9, 46.1, 43.0, 42.3, 37.6, 37.5, 36.8, 32.2, 32.1, 32.1, 28.2, 26.1 (x3), 24.8, 20.9, 19.6, 12.7, −4.4 (x2). HRMS (ESI): *m*/*z* calcd for C_29_H_47_O_2_Si [M+H]^+^: 455.3340, found 455.3350.

#### 3.1.7. General Procedure for the Synthesis of Ynones (**20b**–**i**)

To a solution of an appropriate terminal alkyne **19b**–**i** (2.5 eq) in dry THF (1.5M), n-BuLi (2.3M in hexane, 2.5 eq) was added dropwise at −78 °C. The resulting solution was stirred at −50 °C for 40 min, then a 2M solution of the Weinreb amide **14** (1 eq) in THF was added dropwise at −78 °C. The reaction mixture was slowly warmed to room temperature over 0.5 h and stirred for additional 1–1.5 h (monitored by TLC). Upon completion of the reaction, the mixture was quenched with saturated aq. NH_4_Cl and extracted with EtOAc. The organic layers were washed with water and saturated NaCl, dried over Na_2_SO_4_, and concentrated and purified by silica gel chromatography (PE:EtOAc), thus affording ynones (**20b**–**i**).

##### 1-((17*R*)-3β-((*tert*-Butyldimethylsilyl)oxy)-androst-5-en-17-yl)-5-methylhex-3-yn-2-one (**20b**)

The title compound **20b** (130 mg) was prepared as a white solid in 85% yield using 3-methylbut-1-yne (**19b**) as an alkyne. ^1^H NMR (500 MHz, CDCl_3_) δ 5.31 (dt, *J* = 5.5, 2.1 Hz, 1H), 3.47 (tt, *J* = 11.0, 4.7 Hz, 1H), 2.71 (hept, *J* = 7.0 Hz, 1H), 2.60 (dd, *J* = 15.3, 4.3 Hz, 1H), 2.35 (dd, *J* = 15.3, 9.7 Hz, 1H), 2.26 (tq, *J* = 11.1, 2.8 Hz, 1H), 2.17 (ddd, *J* = 13.3, 5.0, 2.4 Hz, 1H), 1.24 (s, 3H), 1.22 (s, 3H), 1.00 (s, 3H), 0.88 (s, 9H), 0.60 (s, 3H), 0.05 (s, 3H). ^13^C NMR (126 MHz, CDCl_3_) δ 189.0, 141.8, 121.2, 98.8, 80.5, 72.7, 55.7, 50.6, 47.0, 46.4, 43.0, 42.3, 37.6, 37.5, 36.8, 32.2, 32.1, 32.1, 28.3, 26.1 (x3), 24.9, 22.1 (x2), 20.9, 20.9, 19.6, 18.4, 12.7, −4.4 (x2).

##### 1-((17*R*)-3β-((*tert*-Butyldimethylsilyl)oxy)-androst-5-en-17-yl)-4-cyclopropylbut-3-yn-2-one (**20c**)

The title compound **20c** (235 mg) was prepared as a white solid in 78% yield using ethynylcyclopropane (**19c**) as an alkyne. ^1^H NMR (500 MHz, CDCl_3_) δ 5.32 (dd, *J* = 4.9, 2.5 Hz, 1H), 3.48 (tt, *J* = 11.0, 4.7 Hz, 1H), 2.58 (dd, *J* = 15.2, 4.3 Hz, 1H), 2.37–2.21 (m, 2H), 2.17 (ddd, *J* = 13.4, 5.1, 2.3 Hz, 1H), 1.00 (s, 3H), 0.89 (s, 9H), 0.60 (s, 3H), 0.05 (s, 6H). ^13^C NMR (126 MHz, CDCl_3_) δ 188.6, 141.8, 121.2, 98.5, 72.7, 55.7, 50.6, 46.7, 46.4, 43.0, 42.3, 37.6, 37.5, 36.8, 32.2, 32.2, 32.1, 28.3, 26.1 (x3), 24.9, 20.9, 19.6, 18.4, 12.7, 9.8 (x2), −0.1, −4.4 (x2). HRMS (ESI): *m*/*z* calcd for C_32_H_51_O_2_Si [M+H]^+^: 495.3653, found 495.3665.

##### 1-((17*R*)-3β-((*tert*-Butyldimethylsilyl)oxy)-androst-5-en-17-yl)oct-3-yn-2-one (**20d**)

The title compound **20d** (290 mg) was prepared as a white solid in 89% yield using hex-1-yne **3-2c** as an alkyne. ^1^H NMR (500 MHz, CDCl_3_) δ 5.31 (dt, *J* = 4.9, 2.0 Hz, 1H), 3.48 (tt, *J* = 11.0, 4.7 Hz, 1H), 2.60 (dd, *J* = 15.2, 4.2 Hz, 1H), 2.39–2.31 (m, 3H), 2.26 (tq, *J* = 11.2, 2.8 Hz, 1H), 2.17 (ddd, *J* = 13.3, 5.0, 2.3 Hz, 1H), 1.00 (s, 3H), 0.93 (t, *J* = 7.4 Hz, 3H), 0.88 (s, 9H), 0.60 (s, 3H), 0.05 (s, 6H). ^13^C NMR (126 MHz, CDCl_3_) δ 188.8, 141.7, 121.1, 94.2, 81.3, 72.6, 55.6, 50.5, 46.8, 46.3, 42.9, 42.1, 37.5, 37.4, 36.7, 32.1, 32.0, 32.0, 29.8, 28.1, 26.0 (x3), 24.7, 22.0, 20.8, 19.5, 18.7, 18.3, 13.5, 12.6, −4.5 (x2). HRMS (ESI): *m*/*z* calcd for C_33_H_55_O_2_Si [M+H]^+^: 511.3966, found 511.3979.

##### 1-((17*R*)-3β-((*tert*-Butyldimethylsilyl)oxy)-androst-5-en-17-yl)-4-phenylbut-3-yn-2-one (**20e**)

The title compound **20e** (850 mg) was prepared as a white solid in 89% yield using ethynylbenzene (**19e**) as an alkyne. ^1^H NMR (500 MHz, CDCl_3_) δ 7.57 (dt, *J* = 6.9, 1.4 Hz, 2H), 7.49–7.42 (m, 1H), 7.42–7.34 (m, 2H), 5.32 (dd, *J* = 4.8, 2.4 Hz, 1H), 3.48 (tt, *J* = 11.0, 4.7 Hz, 1H), 2.75 (dd, *J* = 15.2, 4.0 Hz, 1H), 2.49 (dd, *J* = 15.1, 9.4 Hz, 1H), 2.27 (ddt, *J* = 13.5, 11.0, 2.7 Hz, 1H), 2.17 (ddd, *J* = 13.3, 5.0, 2.3 Hz, 1H), 1.01 (s, 3H), 0.89 (s, 9H), 0.65 (s, 3H), 0.06 (s, 6H). ^13^C NMR (126 MHz, CDCl_3_) δ 188.6, 141.8, 133.2 (x2), 130.8, 128.8 (x2), 121.2, 120.3, 90.6, 88.4, 72.7, 55.7, 50.6, 47.0, 46.5, 43.0, 42.4, 37.6 (x2), 36.8, 32.2, 32.2, 32.1, 28.3, 26.1 (x3), 24.9, 21.0, 19.6, 18.4, 12.8, −4.4 (x2).

##### 1-((17*R*)-3β-((*tert*-Butyldimethylsilyl)oxy)-androst-5-en-17-yl)-4-(pyridin-3-yl)but-3-yn-2-one (**20f**)

The title compound **20f** (360 mg) was prepared as a white solid in 90% yield using 3-ethynylpyridine (**19f**) as an alkyne. ^1^H NMR (500 MHz, CDCl_3_) δ 8.82–8.76 (m, 1H), 8.66 (dd, *J* = 5.0, 1.6 Hz, 1H), 7.85 (dt, *J* = 7.9, 1.9 Hz, 1H), 7.33 (ddd, *J* = 7.9, 5.0, 0.9 Hz, 1H), 5.32 (dt, *J* = 4.5, 2.0 Hz, 1H), 3.48 (tt, *J* = 11.0, 4.7 Hz, 1H), 2.76 (dd, *J* = 15.4, 4.1 Hz, 1H), 2.51 (dd, *J* = 15.4, 9.4 Hz, 1H), 2.26 (tq, *J* = 11.2, 2.6 Hz, 1H), 2.17 (ddd, *J* = 13.3, 5.0, 2.3 Hz, 1H), 2.05–1.92 (m, 3H), 1.81 (dt, *J* = 13.3, 3.5 Hz, 1H), 1.01 (s, 3H), 0.88 (s, 8H), 0.65 (s, 3H), 0.05 (s, 6H). ^13^C NMR (126 MHz, CDCl_3_) δ 188.1, 153.4, 150.8, 141.8, 139.9, 123.4, 121.1, 117.6, 90.8, 86.5, 72.7, 55.7, 50.6, 47.0, 46.4, 43.0, 42.4, 37.6 (x2), 36.8, 32.2, 32.2, 32.1, 28.3, 26.1 (x3), 24.9, 20.9, 19.6, 18.4, 12.8, −4.4 (x2).

##### 1-((17*R*)-3β-((*tert*-Butyldimethylsilyl)oxy)-androst-5-en-17-yl)-4-(2-fluorophenyl)but-3-yn-2-one (**20g**)

The title compound **20g** (160 mg) was prepared as a white solid in 85% yield using 1-ethynyl-2-fluorobenzene (**19g**) as an alkyne. ^1^H NMR (500 MHz, CDCl_3_) δ 7.55 (ddd, *J* = 7.7, 6.8, 1.8 Hz, 1H), 7.44 (dddd, *J* = 8.4, 7.3, 5.3, 1.8 Hz, 1H), 7.20–7.10 (m, 2H), 5.32 (dt, *J* = 5.5, 2.1 Hz, 1H), 3.48 (tt, *J* = 10.9, 4.7 Hz, 1H), 2.76 (dd, *J* = 15.1, 3.9 Hz, 1H), 2.50 (dd, *J* = 15.3, 9.1 Hz, 1H), 2.32–2.22 (m, 1H), 2.17 (ddd, *J* = 13.3, 5.0, 2.3 Hz, 1H), 2.07–1.95 (m, 3H), 1.81 (dt, *J* = 13.3, 3.6 Hz, 1H), 1.77–1.64 (m, 3H), 1.01 (s, 3H), 0.89 (s, 9H), 0.65 (s, 3H), 0.06 (s, 6H). ^13^C NMR (126 MHz, CDCl_3_) δ 188.4, 163.7 (d, *J* = 255.3 Hz), 141.8, 134.8, 132.8 (d, *J* = 8.0 Hz), 124.5 (d, *J* = 3.6 Hz), 121.2, 116.0 (d, *J* = 20.4 Hz), 92.7, 83.9, 72.8, 55.7, 50.6, 47.1, 46.5, 43.0, 42.4, 37.6, 37.5, 36.9, 32.2, 32.2, 32.1, 28.3, 26.1 (x3), 24.9, 21.0, 19.6, 18.4, 12.8, −4.4 (x2).

##### 1-((17*R*)-3β-((*tert*-Butyldimethylsilyl)oxy)-androst-5-en-17-yl)-5-methyl-5-((tetrahydro-2*H*-pyran-2-yl)oxy)hex-3-yn-2-one (**20h**)

The title compound **20h** (570 mg) was prepared as a white solid in 83% yield using 2-((2-methylbut-3-yn-2-yl)oxy)tetrahydro-2*H*-pyran (**19h**) as an alkyne. ^1^H NMR (500 MHz, CDCl_3_) δ 5.31 (dt, *J* = 4.4, 2.0 Hz, 1H), 5.04–4.96 (m, 1H), 4.00–3.90 (m, 1H), 3.54–3.43 (m, 2H), 2.62 (dd, *J* = 15.3, 4.3 Hz, 1H), 2.38 (dd, *J* = 9.7, 2.3 Hz, 1H), 2.35 (dd, *J* = 9.6, 2.3 Hz, 0H), 2.26 (tq, *J* = 11.2, 2.8 Hz, 1H), 2.16 (ddd, *J* = 13.4, 5.0, 2.3 Hz, 1H), 1.58 (s, 3H), 1.54 (s, 6H), 1.00 (s, 3H), 0.88 (s, 9H), 0.61 (s, 3H), 0.05 (s, 6H). ^13^C NMR (126 MHz, CDCl_3_) δ 188.4, 141.8, 121.1, 96.5, 94.1, 83.3, 72.7, 70.8, 63.5, 55.7, 50.6, 47.0, 46.4, 43.0, 42.3, 37.6, 36.8, 32.2, 32.1, 32.1, 32.0, 29.8, 29.4, 28.2, 26.1, 25.4, 24.8, 20.9, 20.4, 19.6, 18.4, 12.7, −4.4.

##### 1-((17*R*)-3β-((*tert*-Butyldimethylsilyl)oxy)-androst-5-en-17-yl)-5-((tetrahydro-2*H*-pyran-2-yl)oxy)pent-3-yn-2-one (**20i**)

The title compound **20i** (540 mg) was prepared as a white solid in 84% yield using 2-(prop-2-yn-1-yloxy)tetrahydro-2*H*-pyran (**19i**) as an alkyne. ^1^H NMR (500 MHz, CDCl_3_) δ 5.31 (dt, *J* = 4.8, 2.0 Hz, 1H), 4.81 (t, *J* = 3.4 Hz, 1H), 4.42 (s, 2H), 3.83 (ddd, *J* = 11.9, 9.3, 3.0 Hz, 1H), 3.55 (dtd, *J* = 11.3, 4.3, 1.5 Hz, 1H), 3.48 (tt, *J* = 11.0, 4.7 Hz, 1H), 2.64 (dd, *J* = 15.4, 4.2 Hz, 1H), 2.39 (dd, *J* = 15.4, 9.6 Hz, 1H), 2.26 (tq, *J* = 11.1, 2.8 Hz, 1H), 2.17 (ddd, *J* = 13.3, 5.0, 2.3 Hz, 1H), 1.00 (s, 3H), 0.88 (s, 9H), 0.60 (s, 3H), 0.05 (s, 6H). ^13^C NMR (126 MHz, CDCl_3_) δ 188.0, 141.8, 121.1, 97.3, 87.9, 85.3, 72.7, 62.2, 55.7, 54.0, 50.6, 46.8, 46.2, 43.0, 42.3, 37.6, 37.5, 36.8, 32.2, 32.1, 32.1, 30.3, 28.2, 26.1 (x3), 25.4, 24.8, 20.9, 19.6, 19.0, 18.4, 12.8, −4.4 (x2). HRMS (ESI): *m*/*z* calcd for C_35_H_57_O_4_Si [M+H]^+^: 569.4021, found 569.4028.

#### 3.1.8. General Procedure for the Synthesis of Hydroxyisoxazolines (**22a**–**i**)

An aqueous 4M solution of hydroxylamine hydrochloride (2 equiv.) and NaHCO_3_ (2 equiv.) was stirred at room temperature until gas evolution ceased (30 min). Then, a 0.4M THF solution of an appropriate ynone, **20a**–**i** (1 eq.), was added. The biphasic mixture was stirred overnight at room temperature, then partitioned between a saturated aqueous NaCl and EtOAc. The aqueous phase was further extracted with EtOAc. The combined organic washings were dried over Na_2_SO_4_, and the solvent was concentrated under reduced pressure. The residue was purified by silica gel chromatography (PE:EtOAc) to give hydroxyisoxazolines **22a**–**i**.

##### 5-(((17*R*)-3β-((*tert*-Butyldimethylsilyl)oxy)-androst-5-en-17-yl)-4,5-dihydroisoxazol-5-ol (**22a**)

The title compound **22a** (325 mg) was prepared as a white solid in 88% yield from ynone **20a**. ^1^H NMR (500 MHz, CDCl_3_) δ 7.22 (d, *J* = 5.6 Hz, 1H), 5.32 (dd, *J* = 5.1, 2.6 Hz, 1H), 3.49 (ddq, *J* = 15.7, 10.7, 4.7 Hz, 1H), 2.97–2.83 (m, 2H), 2.31–2.22 (m, 1H), 2.17 (ddd, *J* = 13.3, 5.1, 2.3 Hz, 1H), 1.01 (s, 3H), 0.89 (s, 9H), 0.60 (s, 3H), 0.05 (s, 6H). ^13^C NMR (126 MHz, CDCl_3_) δ 147.5, 147.1, 141.8, 121.2, 107.2, 106.8, 72.7, 55.6, 55.5, 50.7, 46.9, 46.6, 46.0, 45.7, 43.0, 38.9, 38.5, 37.6, 37.3, 36.8, 32.2, 32.1, 32.1, 29.8, 29.2, 26.1 (x3), 25.2, 20.9, 19.6, 18.4, 12.5, 12.4, −4.4 (x2). HRMS (ESI): *m*/*z* calcd for C_29_H_50_NO_3_Si [M-H_2_O+H]^+^: 488.3554, found 488.3564.

##### 5-(((17*R*)-3β-((*tert*-Butyldimethylsilyl)oxy)-androst-5-en-17-yl)-3-isopropyl-4,5-dihydroisoxazol-5-ol (**22b**)

The title compound **22b** (170 mg) was prepared as a white solid in 84% yield from ynone **20b**. ^1^H NMR (500 MHz, CDCl_3_) δ 5.31 (d, *J* = 5.4 Hz, 1H), 3.47 (tt, *J* = 10.7, 4.7 Hz, 1H), 2.89–2.66 (m, 3H), 2.26 (ddd, *J* = 13.8, 10.9, 2.8 Hz, 1H), 2.17 (ddd, *J* = 13.3, 5.0, 2.3 Hz, 1H), 1.18 (s, 3H), 1.17 (s, 3H), 1.00 (s, 3H), 0.88 (s, 9H), 0.59 (s, 3H), 0.05 (s, 6H). ^13^C NMR (126 MHz, CDCl_3_) δ 165.1, 164.7, 141.8, 121.2, 121.2, 108.4, 107.9, 72.7, 55.6, 55.5, 50.7, 47.1, 46.6, 45.5, 45.3, 43.0, 42.7, 42.6, 39.1, 38.7, 37.6, 37.3, 36.8, 32.2, 32.1, 32.1, 29.8, 29.2, 28.4, 26.1 (x3), 25.3, 25.2, 20.9, 20.3, 20.0, 19.6, 18.4, 12.5, 12.4, −4.4 (x2).

##### 5-(((17*R*)-3β-((*tert*-Butyldimethylsilyl)oxy)-androst-5-en-17-yl)-3-cyclopropyl-4,5-dihydroisoxazol-5-ol (**22c**)

The title compound **22c** (130 mg) was prepared as a white solid in 55% yield from ynone **20c**. ^1^H NMR (500 MHz, CDCl_3_) δ 5.32 (dd, *J* = 5.3, 2.5 Hz, 1H), 3.48 (tt, *J* = 10.5, 4.7 Hz, 1H), 2.74 (dd, *J* = 17.2, 9.0 Hz, 1H), 2.61 (dd, *J* = 28.2, 17.2 Hz, 1H), 2.26 (tq, *J* = 11.2, 2.8 Hz, 1H), 2.17 (ddd, *J* = 13.3, 5.0, 2.2 Hz, 1H), 1.00 (s, 3H), 0.88 (s, 9H), 0.77–0.74 (m, 2H), 0.58 (s, 3H), 0.05 (s, 6H). ^13^C NMR (126 MHz, CDCl_3_) δ 162.5, 162.1, 141.8, 121.2, 108.3, 107.9, 72.7, 55.6, 55.5, 50.7, 47.0, 46.6, 45.6, 45.3, 43.0, 42.7, 42.6, 39.0, 38.6, 37.6, 37.3, 37.3, 36.8, 32.2, 32.1, 32.1, 29.8, 29.2, 26.1 (x3), 25.3, 25.2, 20.9, 19.6, 18.4, 12.5, 12.4, 9.5, 6.8, 6.8, 6.0, −4.4 (x2). HRMS (ESI): *m*/*z* calcd for C_32_H_52_NO_2_Si [M-H_2_O+H]^+^: 510.3762, found 510.3776.

##### 3-Butyl-5-(((17*R*)-3β-((*tert*-butyldimethylsilyl)oxy)-androst-5-en-17-yl)methyl)-4,5-dihydroisoxazol-5-ol (**22d**)

The title compound **22d** (203 mg) was prepared as a white solid in 68% yield from ynone **20d**. ^1^H NMR (500 MHz, CDCl_3_) δ 5.31 (dt, *J* = 4.9, 2.2 Hz, 1H), 3.48 (tt, *J* = 10.9, 4.7 Hz, 1H), 2.90–2.72 (m, 2H), 2.36 (ddd, *J* = 11.5, 6.6, 3.0 Hz, 2H), 2.26 (tq, *J* = 11.1, 2.7 Hz, 1H), 2.17 (ddd, *J* = 13.3, 5.0, 2.2 Hz, 1H), 1.00 (s, 3H), 0.93 (t, *J* = 7.4 Hz, 3H), 0.88 (s, 9H), 0.59 (s, 3H), 0.05 (s, 6H). ^13^C NMR (126 MHz, CDCl_3_) δ 160.7, 160.3, 141.8, 121.2, 121.2, 108.4, 107.9, 72.7, 55.6, 55.6, 50.7, 47.5, 47.2, 47.1, 46.6, 43.0, 42.7, 42.6, 39.1, 38.7, 37.6, 37.3, 36.8, 32.2, 32.1, 32.1, 29.9, 29.2, 28.6, 27.9, 26.1 (x3), 25.3, 25.2, 22.5, 20.9, 19.6, 18.4, 13.9, 12.5, 12.4, −4.4 (x2).

##### 5-(((17*R*)-3β-((*tert*-Butyldimethylsilyl)oxy)-androst-5-en-17-yl)methyl)-3-phenyl-4,5-dihydroisoxazol-5-ol (**22e**)

The title compound **22e** (220 mg) was prepared as a white solid in 61% yield from ynone **20e**. ^1^H NMR (500 MHz, CDCl_3_) δ 7.65 (ddd, *J* = 6.0, 3.1, 1.4 Hz, 2H), 7.45–7.29 (m, 3H), 5.36–5.26 (m, 1H), 3.49 (tt, *J* = 10.9, 4.6 Hz, 1H), 1.01 (s, 3H), 0.89 (d, *J* = 1.4 Hz, 9H), 0.61 (s, 3H), 0.06 (s, 6H). ^13^C NMR (126 MHz, CDCl_3_) δ 157.8, 157.4, 141.8, 132.2, 132.0, 130.3, 129.8, 128.8, 128.5, 128.5, 126.8, 126.8, 121.2, 109.6, 109.2, 72.7, 55.6, 55.5, 50.7, 50.6, 46.9, 46.6, 45.4, 45.1, 42.9, 42.7, 42.6, 39.2, 38.7, 37.6, 37.3, 36.8, 32.2, 32.2, 32.1, 32.1, 29.9, 29.8, 29.3, 26.1 (x3), 25.3, 25.2, 20.9, 19.6, 18.4, 12.5, 12.4, −4.4 (x2). HRMS (ESI): *m*/*z* calcd for C_35_H_52_NO_2_Si [M-H_2_O+H]^+^: 546.3762, found 546.3773.

##### 5-(((17*R*)-3β-((*tert*-Butyldimethylsilyl)oxy)-androst-5-en-17-yl)-3-(pyridin-3-yl)-4,5-dihydroisoxazol-5-ol (**22f**)

The title compound **22f** (200 mg) was prepared as a white solid in 56% yield from ynone **20f**. ^1^H NMR (500 MHz, CDCl_3_) δ 8.76 (t, *J* = 2.8 Hz, 1H), 8.61 (dd, *J* = 5.0, 1.6 Hz, 1H), 8.06–7.99 (m, 1H), 7.36–7.30 (m, 1H), 5.32 (dt, *J* = 5.2, 1.7 Hz, 1H), 3.89 (s, 1H), 3.48 (tt, *J* = 10.6, 4.7 Hz, 1H), 3.31–3.19 (m, 2H), 2.27 (tq, *J* = 11.2, 2.7 Hz, 1H), 2.17 (ddd, *J* = 15.1, 6.0, 1.9 Hz, 2H), 2.04–1.92 (m, 1H), 1.01 (s, 3H), 0.88 (s, 9H), 0.62 (s, 3H), 0.05 (s, 6H). ^13^C NMR (126 MHz, CDCl_3_) δ 155.2, 154.8, 151.0, 147.7, 141.8, 133.9, 133.8, 126.2, 123.8, 121.2, 110.1, 109.7, 72.7, 55.6, 55.5, 50.7, 46.9, 46.6, 44.8, 44.4, 43.0, 42.8, 42.7, 39.2, 38.7, 37.6, 37.4, 36.8, 32.2, 32.1, 29.9, 29.3, 26.1 (x3), 25.3, 25.2, 20.9, 19.6, 18.4, 12.5, 12.5, −4.4 (x2).

##### 5-(((17*R*)-3β-((*tert*-Butyldimethylsilyl)oxy)-androst-5-en-17-yl)-3-(2-fluorophenyl)-4,5-dihydroisoxazol-5-ol (**22g**)

The title compound **22g** (60 mg) was prepared as a white solid in 68% yield from ynone **20g**. ^1^H NMR (500 MHz, CDCl_3_) δ 7.88 (tt, *J* = 7.6, 1.6 Hz, 1H), 7.39 (dddd, *J* = 8.6, 7.1, 5.1, 1.8 Hz, 1H), 7.18 (td, *J* = 7.6, 1.2 Hz, 1H), 7.15–7.06 (m, 1H), 5.32 (dt, *J* = 5.4, 1.9 Hz, 1H), 3.48 (td, *J* = 10.9, 5.1 Hz, 1H), 3.42–3.27 (m, 2H), 2.27 (tq, *J* = 11.0, 2.7 Hz, 1H), 1.01 (s, 3H), 0.89 (s, 9H), 0.63–0.61 (m, 3H), 0.06 (s, 6H). ^13^C NMR (126 MHz, CDCl_3_) δ 160.5 (d, *J* = 252.3 Hz), 154.2 (d, *J* = 48.4 Hz), 141.8, 132.0 (d, *J* = 8.8 Hz), 129.0, 124.6, 121.2, 117.9 (d, *J* = 11.6 Hz), 116.5 (d, *J* = 21.9 Hz), 109.4 (d, *J* = 52.4 Hz), 72.8, 55.6, 55.5, 50.7, 47.3, 47.0, 46.9, 46.6, 43.0, 42.8, 42.7, 39.1, 38.7, 37.6, 37.3, 36.9, 32.2, 32.1, 30.2, 29.9, 29.3, 26.1 (x3), 25.6, 25.3, 25.2, 20.9, 19.6, 18.4, 12.5, 12.5, −4.4 (x2). HRMS (ESI): *m*/*z* calcd for C_35_H_51_FNO_2_Si [M-H_2_O+H]^+^: 564.3668, found 564.3680.

##### 5-(((17*R*)-3β-((*tert*-Butyldimethylsilyl)oxy)-androst-5-en-17-yl)-3-(2-((tetrahydro-2*H*-pyran-2-yl)oxy)propan-2-yl)-4,5-dihydroisoxazol-5-ol (**22h**)

The title compound **22h** (520 mg) was prepared as a white solid in 83% yield from ynone **20h**. ^1^H NMR (500 MHz, CDCl_3_) δ 5.33–5.30 (m, 1H), 5.29 (s, 1H), 5.25 (s, 0.5H), 5.20 (s, 0.5H), 4.83 (dt, *J* = 5.9, 3.0 Hz, 1H), 3.89 (ddd, *J* = 11.2, 7.4, 3.0 Hz, 1H), 3.48 (ddt, *J* = 15.7, 10.9, 4.6 Hz, 2H), 2.91–2.74 (m, 2H), 2.26 (tq, *J* = 11.0, 2.7 Hz, 1H), 2.21–2.10 (m, 2H), 1.38 (s, 3H), 1.00 (s, 3H), 0.88 (m, 12H), 0.59 (s, 3H), 0.05 (s, 6H). ^13^C NMR (126 MHz, CDCl_3_) δ 162.6, 162.2, 141.8, 141.8, 121.3, 121.2, 109.5, 109.2, 94.3, 94.2, 73.8, 73.8, 72.8, 63.4, 63.2, 55.7, 55.6, 50.8, 47.0, 46.6, 43.9, 43.5, 43.0, 42.6, 42.5, 37.6, 37.4, 37.3, 36.9, 36.9, 36.6, 32.2, 32.2, 32.1, 32.0, 31.9, 29.6, 29.1, 26.9, 26.1 (x3), 25.3, 25.2, 22.7, 20.9, 19.9, 19.8, 19.6, 18.4, 12.5, 12.5, −4.4 (x2). HRMS (ESI): *m*/*z* calcd for C_32_H_54_NO_3_Si [M-OTHP]^+^: 528.3867, found 528.3856.

##### 5-(((17*R*)-3β-((*tert*-Butyldimethylsilyl)oxy)-androst-5-en-17-yl)methyl)-3-(((tetrahydro-2*H*-pyran-2-yl)oxy)methyl)-4,5-dihydroisoxazol-5-ol (**22i**)

The title compound **22i** (630 mg) was prepared as a white solid in 86% yield from ynone **20i**. ^1^H NMR (500 MHz, CDCl_3_) δ 5.32 (dt, *J* = 5.0, 2.1 Hz, 1H), 4.65 (ddt, *J* = 7.3, 5.0, 2.7 Hz, 1H), 4.51–4.40 (m, 1H), 4.33 (ddd, *J* = 12.7, 8.5, 3.8 Hz, 1H), 3.93–3.79 (m, 1H), 3.58–3.43 (m, 2H), 3.03–2.76 (m, 2H), 2.27 (tq, *J* = 11.3, 2.8 Hz, 1H), 1.00 (s, 3H), 0.88 (s, 9H), 0.60 (s, 3H), 0.05 (s, 6H). ^13^C NMR (126 MHz, CDCl_3_) δ 158.3, 141.8, 121.2, 109.4, 109.3, 109.0, 108.8, 99.8, 99.6, 98.8, 63.4, 63.3, 63.3, 63.1, 62.7, 62.2, 55.6, 50.7, 46.9, 46.6, 45.9, 45.6, 45.5, 45.2, 43.0, 42.6, 39.1, 38.6, 38.3, 37.8, 37.6, 37.3, 36.8, 32.2, 32.1, 32.1, 30.8, 30.5, 29.8, 29.2, 26.1, 25.4, 25.3, 20.9, 19.8, 19.7, 19.6, 19.5, 18.4, 12.5, 12.4, −4.4. HRMS (ESI): *m*/*z* calcd for C_35_H_60_NO_5_Si [M+H]^+^: 602.4235, found 602.4242; calcd for C_35_H_58_NO_4_Si [M-H_2_O+H]^+^: 584.4130, found 584.4136.

#### 3.1.9. General Procedure for the Synthesis of Isoxazoles (**23a**–**i**)

CDI (1.7 eq.) was added to a 0.4M solution of dihydroisoxazololes **22a**–**i** (1 eq.) in dry CH_2_Cl_2_, and the mixture was stirred at room temperature for 16 h. Upon completion of the reaction, the mixture was concentrated under reduced pressure, and the residue was chromatographed on silica gel (PE:EtOAc) to afford isoxazoles **23a**–**i**.

##### 5-(((17*R*)-3β-((*tert*-butyldimethylsilyl)oxy)-androst-5-en-17-yl)methyl)isoxazole (**23a**)

The title compound **23a** (110 mg) was prepared as a white solid in 51% yield from hydroxyisoxazoline **22a** together with 4-(3β-((*tert*-butyldimethylsilyl)oxy)-androst-5-en-17-yl)-3-oxobutanenitrile (**31**) (85 mg, 40%) isolated as a slightly yellow solid.

Isoxazole **23a**: ^1^H NMR (500 MHz, CDCl_3_) δ 8.12 (d, *J* = 1.7 Hz, 1H), 5.96 (d, *J* = 1.7 Hz, 1H), 5.31 (dt, *J* = 5.5, 2.0 Hz, 1H), 3.48 (tt, *J* = 10.9, 4.7 Hz, 1H), 2.85 (dd, *J* = 15.0, 5.0 Hz, 1H), 2.59 (dd, *J* = 15.0, 9.7 Hz, 1H), 2.26 (ddd, *J* = 13.6, 10.8, 2.7 Hz, 1H), 2.17 (ddd, *J* = 13.3, 5.0, 2.4 Hz, 1H), 1.99 (dtd, *J* = 16.9, 5.0, 2.6 Hz, 1H), 1.00 (s, 3H), 0.89 (s, 9H), 0.69 (s, 3H), 0.05 (s, 6H). ^13^C NMR (126 MHz, CDCl_3_) δ 173.0, 150.3, 141.8, 121.1, 100.3, 72.7, 55.9, 50.6, 49.1, 43.0, 42.3, 37.6, 37.5, 36.8, 32.2, 32.1, 32.1, 28.6, 27.6, 26.1 (x3), 24.7, 20.9, 19.6, 18.4, 12.4, −4.4 (x2). HRMS (ESI): *m*/*z* calcd for C_29_H_48_NO_2_Si [M+H]^+^: 470.3449, found 470.3460.

β-Oxonitrile **31**: ^1^H NMR (500 MHz, CDCl_3_) δ 5.31 (dq, *J* = 5.5, 3.3, 2.7 Hz, 1H), 3.52–3.41 (m, 3H), 2.69 (dd, *J* = 16.4, 3.9 Hz, 1H), 2.41 (dd, *J* = 16.5, 10.1 Hz, 1H), 2.26 (tq, *J* = 11.1, 2.8 Hz, 1H), 2.17 (ddd, *J* = 13.3, 5.0, 2.3 Hz, 1H), 1.00 (s, 3H), 0.89 (s, 9H), 0.61 (s, 3H), 0.05 (s, 6H). ^13^C NMR (126 MHz, CDCl_3_) δ 197.8, 141.7, 121.1, 114.0, 72.7, 55.6, 50.5, 45.6, 43.4, 42.9, 42.2, 37.6, 37.4, 36.8, 32.3, 32.2, 32.1, 32.0, 28.4, 26.1, 24.8, 20.9, 19.6, 18.4, 12.8, −4.4.

##### 5-(((17*R*)-3β-((*tert*-Butyldimethylsilyl)oxy)-androst-5-en-17-yl)methyl)-3-isopropylisoxazole (**23b**)

The title compound **23b** (45 mg) was prepared as a white solid in 27% yield from hydroxyisoxazoline **22b** together with (*E*)-4-amino-1-((17*R*)-3β-((*tert*-butyldimethylsilyl)oxy)-androst-5-en-17-yl)-5-methyl-2-oxohex-3-en-3-yl 1*H*-imidazole-1-carboxylate (**26b**) (103 mg, 50%) isolated as a white solid.

Isoxazole **23b**: ^1^H NMR (500 MHz, CDCl_3_) δ 5.81 (s, 1H), 5.31 (dt, *J* = 5.5, 2.0 Hz, 1H), 3.00 (hept, *J* = 6.9 Hz, 1H), 2.78 (dd, *J* = 15.1, 4.9 Hz, 1H), 2.50 (dd, *J* = 15.1, 10.0 Hz, 1H), 2.26 (tq, *J* = 11.1, 2.7 Hz, 1H), 2.16 (ddd, *J* = 13.4, 5.0, 2.3 Hz, 1H), 1.26 (s, 3H), 1.25 (s, 3H), 1.00 (s, 3H), 0.88 (s, 9H), 0.67 (s, 3H), 0.05 (s, 6H). ^13^C NMR (126 MHz, CDCl_3_) δ 173.2, 169.4, 141.8, 121.1, 99.0, 56.0, 50.6, 49.1, 43.0, 42.3, 37.6, 37.5, 36.8, 32.2, 32.1, 32.1, 28.6, 27.7, 26.6, 26.1 (x3), 24.7, 21.9 (x2), 20.9, 19.6, 18.4, 12.4, −4.4 (x2).

Imidazolecarboxylate **26b**: ^1^H NMR (500 MHz, CDCl_3_) δ 8.21 (d, *J* = 1.1 Hz, 1H), 7.56 (t, *J* = 1.4 Hz, 1H), 7.20 (t, *J* = 1.2 Hz, 1H), 5.31 (dt, *J* = 5.6, 2.0 Hz, 1H), 3.82 (hept, *J* = 7.0 Hz, 1H), 3.47 (tt, *J* = 10.9, 4.7 Hz, 1H), 3.01 (dd, *J* = 15.7, 3.6 Hz, 1H), 2.72 (dd, *J* = 15.7, 9.1 Hz, 1H), 2.26 (ddd, *J* = 13.7, 10.8, 2.8 Hz, 2H), 2.17 (ddd, *J* = 13.3, 5.0, 2.3 Hz, 1H), 1.32 (d, *J* = 2.1 Hz, 3H), 1.31 (d, *J* = 2.2 Hz, 3H), 1.01 (s, 3H), 0.88 (s, 9H), 0.69 (s, 3H), 0.05 (s, 6H). ^13^C NMR (126 MHz, CDCl_3_) δ 197.4, 160.4, 147.7, 141.8, 135.3, 132.7, 131.0, 121.2, 116.6, 72.7, 55.8, 50.6, 46.1, 43.0, 42.3, 41.4, 37.6, 37.5, 36.8, 32.2, 32.2, 32.1, 28.4, 26.3, 26.1 (x3), 24.9, 21.0, 20.6, 20.5, 19.6, 18.4, 12.8, −4.4 (x2).

##### 5-(((17*R*)-3β-((*tert*-Butyldimethylsilyl)oxy)-androst-5-en-17-yl)methyl)-3-cyclopropylisoxazole (**23c**)

The title compound **23c** (11 mg) was prepared as a white solid in 9% yield from hydroxyisoxazoline **22c** together with (*E*)-1-amino-4-((17*R*)-3β-((*tert*-butyldimethylsilyl)oxy)-androst-5-en-17-yl)-1-cyclopropyl-3-oxobut-1-en-2-yl 1*H*-imidazole-1-carboxylate **26c** (35 mg, 44%) isolated as a white solid.

Isoxazole **23c**: ^1^H NMR (500 MHz, CDCl_3_) δ 5.60 (s, 1H), 5.31 (dt, *J* = 5.0, 2.0 Hz, 1H), 3.47 (tt, *J* = 11.0, 4.7 Hz, 1H), 2.75 (dd, *J* = 15.0, 5.0 Hz, 1H), 2.47 (dd, *J* = 15.0, 10.0 Hz, 1H), 2.31–2.22 (m, 1H), 2.16 (ddd, *J* = 13.4, 5.1, 2.3 Hz, 1H), 1.00 (s, 3H), 0.88 (s, 9H), 0.66 (s, 3H), 0.05 (s, 6H). ^13^C NMR (126 MHz, CDCl_3_) δ 173.3, 166.4, 141.8, 121.1, 98.4, 72.7, 55.9, 50.6, 49.0, 43.0, 42.3, 37.6, 37.5, 36.8, 32.2, 32.1, 32.1, 28.6, 27.7, 26.1 (x3), 24.7, 20.9, 19.6, 18.4, 12.4, 8.0 (x2), 7.5, −4.4 (x2). 

Imidazolecarboxylate **26c**: ^1^H NMR (500 MHz, CDCl_3_) δ 8.10 (s, Im-2, 1H), 7.46 (t, *J* = 1.5, 1H), 7.14 (s, Im-4, 1H), 5.29 (dt, H-6, *J* = 5.6, 2.0 Hz, 1H), 3.45 (tt, H-3, *J* = 10.9, 4.7, 1H), 2.98 (dd, H-20, *J* = 15.6, 3.4 Hz, 1H), 2.83 (tt, H-25, *J* = 8.5, 5.2 Hz, 1H), 2.70 (dd, H-20, *J* = 15.6, 9.1 Hz, 1H), 2.24 (tq, H-4, *J* = 11.0, 2.6 Hz, 1H), 2.14 (ddd, H-4, *J* = 13.4, 5.1, 2.3 Hz, 1H), 1.96 (m, H-2, 1H), 1.93 (m, H-17, 1H), 1.93 (m, H-16, 1H), 1.80 (m, H-1, 1H), 1.74 (m, H-12, 1H), 1.74 (m, H-1, 1H), 1.67 (m, H-7, 1H), 1.63 (m, H-15, 1H), 1.50 (m, H-2, 1H), 1.44 (m, H-8, 1H), 1.43 (m, H-11, 1H), 1.30 (m, H-16, 1H), 1.18 (m, H-26 and H-27, 2H), 1.15 (m, H-15, 1H), 1.08 (m, H-26 and H-27, 2H), 1.06 (m, H-12, 1H), 1.01 (m, H-14, 1H), 0.98 (s, H-19, 3H), 0.93 (m, H-9, 1H), 0.85 (s, SiCMe_3_, 9H), 0.67 (s, H-18, 3H), 0.02 (s, SiMe_2_, 6H). ^13^C NMR (126 MHz, CDCl_3_) δ 197.1 (s, C-22), 157.2 (s, C-24), 146.4 (s, >C=O), 141.6 (s, C-5), 135.1 (s, Im-2), 134.2 (s, C-23), 130.9 (s, Im-4), 121.1 (d, C-6), 116.5 (s, Im-5), 72.6 (d, C-3), 55.7 (d, C-14), 50.6 (d, C-9), 46.0 (d, C-17), 42.9 (t, C-4), 42.3 (s, C-13), 41.1 (d, C-20), 37.5 (t, C-12), 37.4 (t, C-1), 36.8 (s, C-10), 32.2 (t, C-7), 32.1 (d, C-8), 32.0 (t, C-2), 28.4 (t, C-16), 26.0 (s, SiCMe_3_, x3), 24.8 (t, C-15), 20.9 (t, C-11), 19.5 (s, C-19), 18.3 (s, SiCMe_3_), 12.8 (s, C-18), 9.5 (t, C-26 and C-27, x2), 8.2 (d, C-25), −4.5 (s, SiMe_2_, x2). Selected HMBC correlations are between δ 2.98, 2.70 (H-20), 1.93 (H-17) and 197.1 (C-22), between δ 2.83 (H-25), 1.18, 1.08 (H-26, H-27) and 157.2 (C-24), between δ 2.70 (H-20) and 134.2 (C-23).

##### 3-Butyl-5-(((17*R*)-3β-((*tert*-butyldimethylsilyl)oxy)-androst-5-en-17-yl)methyl)isoxazole (**23d**)

The title compound **23d** (100 mg) was prepared as a white solid in 58% yield from hydroxyisoxazoline **22d**. ^1^H NMR (500 MHz, CDCl_3_) δ 5.79 (s, 1H), 5.31 (dt, *J* = 4.8, 2.1 Hz, 1H), 3.47 (tt, *J* = 11.1, 4.7 Hz, 1H), 2.78 (dd, *J* = 15.0, 5.1 Hz, 1H), 2.61 (t, *J* = 7.7 Hz, 2H), 2.51 (dd, *J* = 15.0, 9.9 Hz, 1H), 2.26 (ddd, *J* = 13.6, 10.9, 2.7 Hz, 1H), 2.17 (ddd, *J* = 13.4, 5.1, 2.3 Hz, 1H), 1.98 (dtd, *J* = 16.9, 5.0, 2.6 Hz, 1H), 1.00 (s, 3H), 0.93 (t, *J* = 7.3 Hz, 4H), 0.88 (s, 9H), 0.67 (s, 3H), 0.05 (s, 6H). ^13^C NMR (126 MHz, CDCl_3_) δ 173.2, 164.1, 141.8, 121.1, 100.7, 72.7, 56.0, 50.6, 49.1, 43.0, 42.3, 37.6, 37.5, 36.8, 32.2, 32.1, 32.1, 30.6, 28.6, 27.7, 26.1 (x3), 25.9, 24.7, 22.4, 20.9, 19.6, 18.4, 13.9, 12.4, −4.4 (x2).

##### 5-(((17*R*)-3β-((*tert*-Butyldimethylsilyl)oxy)-androst-5-en-17-yl)methyl)-3-phenylisoxazole (**23e**)

The title compound **23e** (105 mg) was prepared as a white solid in 61% yield from hydroxyisoxazoline **22e**. ^1^H NMR (500 MHz, CDCl_3_) δ 7.83–7.75 (m, 2H), 7.48–7.38 (m, 3H), 6.28 (s, 1H), 5.32 (dt, *J* = 5.5, 2.0 Hz, 1H), 3.48 (tt, *J* = 11.0, 4.7 Hz, 1H), 2.87 (dd, *J* = 15.0, 5.1 Hz, 1H), 2.61 (dd, *J* = 15.0, 9.7 Hz, 1H), 2.32–2.23 (m, 1H), 2.18 (ddd, *J* = 13.3, 5.0, 2.3 Hz, 1H), 1.01 (s, 3H), 0.89 (s, 9H), 0.71 (s, 3H), 0.06 (s, 6H). ^13^C NMR (126 MHz, CDCl_3_) δ 174.2, 162.4, 141.7, 129.9, 129.6, 128.9 (x2), 126.9 (x2), 121.1, 99.2, 72.7, 55.9, 50.6, 49.1, 43.0, 42.3, 37.5, 37.5, 36.8, 32.2, 32.1, 32.1, 28.6, 27.8, 26.1 (x3), 24.7, 20.9, 19.6, 18.4, 12.4, −4.4 (x2). HRMS (ESI): *m*/*z* calcd for C_35_H_52_NO_2_Si [M+H]^+^: 546.3762, found 546.3767.

##### 5-(((17*R*)-3β-((*tert*-Butyldimethylsilyl)oxy)-androst-5-en-17-yl)methyl)-3-(pyridin-3-yl)isoxazole (**23f**)

The title compound **23f** (54 mg) was prepared as a white solid in 60% yield from hydroxyisoxazoline **22f**. ^1^H NMR (500 MHz, CDCl_3_) δ 8.98 (d, *J* = 2.3 Hz, 1H), 8.66 (dd, *J* = 4.8, 1.7 Hz, 1H), 8.13 (dt, *J* = 7.9, 2.0 Hz, 1H), 7.38 (dd, *J* = 7.9, 4.8 Hz, 1H), 6.33 (s, 1H), 5.32 (dd, *J* = 4.8, 2.4 Hz, 1H), 3.48 (tt, *J* = 11.0, 4.7 Hz, 1H), 2.90 (dd, *J* = 15.1, 5.0 Hz, 1H), 2.63 (dd, *J* = 15.1, 9.7 Hz, 1H), 2.27 (ddd, *J* = 13.6, 11.0, 3.0 Hz, 1H), 2.17 (ddd, *J* = 13.4, 5.1, 2.3 Hz, 1H), 1.01 (s, 3H), 0.88 (s, 9H), 0.71 (s, 2H), 0.05 (s, 6H). ^13^C NMR (126 MHz, CDCl_3_) δ 175.0, 159.9, 150.9, 148.1, 141.8, 134.1, 125.8, 123.9, 121.1, 98.9, 72.7, 56.0, 50.6, 49.1, 43.0, 42.4, 37.6 (x2), 36.8, 32.2, 32.1, 32.1, 28.6, 27.9, 26.1 (x3), 24.7, 20.9, 19.6, 18.4, 12.5, −4.4 (x2). HRMS (ESI): *m*/*z* calcd for C_34_H_51_N_2_O_2_Si [M+H]^+^: 547.3714, found 547.3717.

##### 5-(((17*R*)-3β-((*tert*-Butyldimethylsilyl)oxy)-androst-5-en-17-yl)methyl)-3-(2-fluorophenyl)isoxazole (**23g**)

The title compound **23g** (20 mg) was prepared as a white solid in 71% yield from hydroxyisoxazoline **22g**. ^1^H NMR (500 MHz, CDCl_3_) δ 7.98 (td, *J* = 7.6, 1.8 Hz, 1H), 7.40 (dddd, *J* = 8.4, 7.1, 5.2, 1.8 Hz, 1H), 7.22 (td, *J* = 7.6, 1.2 Hz, 1H), 7.16 (ddd, *J* = 11.1, 8.3, 1.2 Hz, 1H), 6.43 (d, *J* = 3.7 Hz, 1H), 5.32 (dt, *J* = 5.5, 2.0 Hz, 1H), 3.48 (tt, *J* = 11.0, 4.7 Hz, 1H), 2.89 (dd, *J* = 15.1, 5.0 Hz, 1H), 2.63 (dd, *J* = 15.1, 9.7 Hz, 1H), 2.27 (ddd, *J* = 13.6, 10.8, 2.7 Hz, 1H), 2.17 (ddd, *J* = 13.3, 5.0, 2.3 Hz, 1H), 1.99 (dtd, *J* = 16.9, 5.0, 2.6 Hz, 1H), 1.01 (s, 3H), 0.89 (s, 9H), 0.71 (s, 3H), 0.06 (s, 6H). ^13^C NMR (126 MHz, CDCl_3_) δ 174.1, 160.4 (d, *J* = 251.3 Hz), 157.8, 141.8, 131.5 (d, *J* = 8.6 Hz), 129.2 (d, *J* = 3.0 Hz), 124.7 (d, *J* = 3.5 Hz), 121.1, 117.7 (d, *J* = 12.0 Hz), 116.4 (d, *J* = 22.0 Hz), 101.8 (d, *J* = 9.0 Hz), 56.0, 50.6, 49.1, 43.0, 42.3, 37.6, 37.5, 36.8, 32.2, 32.1, 32.1, 28.6, 27.8, 26.1 (x3), 24.7, 20.9, 19.6, 18.4, 12.5, 1.2, −4.4 (x2).

##### 5-(((17*R*)-3β-((*tert*-Butyldimethylsilyl)oxy)-androst-5-en-17-yl)-3-(2-((tetrahydro-2*H*-pyran-2-yl)oxy)propan-2-yl)isoxazole (**23h**)

The title compound **23h** (384 mg) was prepared as a white solid in 78% yield from hydroxyisoxazoline **22h**. ^1^H NMR (500 MHz, CDCl_3_) δ 5.97 (s, 1H), 5.31 (dt, *J* = 5.1, 2.1 Hz, 1H), 4.50 (dt, *J* = 6.1, 2.8 Hz, 1H), 3.91 (dd, *J* = 11.0, 4.9 Hz, 1H), 3.47 (tt, *J* = 8.4, 3.5 Hz, 1H), 3.39 (dddd, *J* = 9.4, 7.0, 4.2, 2.2 Hz, 1H), 2.82–2.76 (m, 1H), 2.54 (ddd, *J* = 15.0, 9.5, 3.6 Hz, 1H), 2.26 (tt, *J* = 14.0, 2.9 Hz, 1H), 2.16 (ddd, *J* = 13.4, 5.0, 2.2 Hz, 1H), 1.98 (dtd, *J* = 16.7, 5.0, 2.5 Hz, 1H), 1.65 (s, 2H), 1.54 (s, 4H), 1.00 (s, 4H), 0.88 (s, 9H), 0.68, 0.67 (s, 3H), 0.05 (s, 6H). ^13^C NMR (126 MHz, CDCl_3_) δ 173.5, 173.4, 168.9, 168.8, 141.8, 121.1, 99.5, 99.4, 95.8, 95.7, 74.4, 72.7, 63.9, 63.9, 56.0, 50.6, 49.2, 48.9, 43.0, 42.3, 37.6, 37.5, 36.8, 32.3, 32.2, 32.1, 32.1, 29.2, 29.1, 28.6, 28.6, 27.8, 27.7, 26.1 (x3), 25.9, 25.8, 25.4, 24.7, 20.9, 19.6, 18.4, 12.4, −4.4 (x2).

##### 5-(((17*R*)-3β-((*tert*-Butyldimethylsilyl)oxy)-androst-5-en-17-yl)-3-(((tetrahydro-2*H*-pyran-2-yl)oxy)methyl)isoxazole (**23i**)

The title compound **23i** (530 mg) was prepared as a white solid in 87% yield from hydroxyisoxazoline **22i**. ^1^H NMR (500 MHz, CDCl_3_) δ 6.03 (s, 1H), 5.31 (dt, *J* = 5.5, 2.0 Hz, 1H), 4.73 (d, *J* = 12.8 Hz, 1H), 4.70 (t, *J* = 3.6 Hz, 1H), 4.57 (d, *J* = 12.8 Hz, 1H), 3.88 (ddd, *J* = 11.4, 8.3, 2.9 Hz, 1H), 3.58–3.52 (m, 2H), 3.47 (tt, *J* = 11.0, 4.7 Hz, 1H), 2.81 (dd, *J* = 15.0, 4.9 Hz, 1H), 2.54 (dd, *J* = 15.1, 10.0 Hz, 1H), 2.26 (tq, *J* = 11.0, 2.7 Hz, 1H), 2.16 (ddd, *J* = 13.3, 5.0, 2.3 Hz, 1H), 1.98 (dtd, *J* = 16.8, 5.0, 2.6 Hz, 1H), 1.00 (s, 3H), 0.88 (s, 9H), 0.68 (s, 3H), 0.05 (s, 6H). ^13^C NMR (126 MHz, CDCl_3_) δ 173.9, 161.6, 141.8, 121.1, 100.6, 100.6, 98.4, 72.7, 62.4, 60.7, 55.9, 50.6, 49.0, 43.0, 42.3, 37.6, 37.5, 36.8, 32.2, 32.1, 32.1, 30.5, 28.6, 27.7, 26.1 (x3), 25.5, 24.7, 20.9, 19.6, 19.4, 18.4, 12.4, −4.4 (x2). HRMS (ESI): *m*/*z* calcd for C_35_H_58_NO_4_Si [M+H]^+^: 584.4130, found 584.4143.

#### 3.1.10. (E)-1-Amino-4-((17*R*)-3β-hydroxy-androst-5-en-17-yl)-1-cyclopropyl-3-oxobut-1-en-2-yl 1H-imidazole-1-carboxylate (**27**)

The title compound 27 (35 mg) was obtained from 26c in a similar manner for the preparation of **24j** as a white solid in 87% yield. ^1^H NMR (500 MHz, CDCl_3_) δ 8.13 (s, 1H), 7.52–7.45 (m, 1H), 7.17 (s, 1H), 5.35 (dt, J = 4.7, 2.0 Hz, 1H), 3.52 (tt, J = 11.1, 4.7 Hz, 1H), 3.00 (dd, J = 15.6, 3.4 Hz, 1H), 2.85 (tt, J = 8.4, 5.2 Hz, 1H), 2.72 (dd, J = 15.6, 9.0 Hz, 1H), 2.34–2.18 (m, 2H), 1.02 (s, 3H), 0.70 (s, 3H). ^13^C NMR (126 MHz, CDCl_3_) δ 197.2, 157.3, 146.5, 141.0, 135.2, 134.3, 131.0, 121.7, 116.6, 71.8, 55.7, 50.6, 46.1, 42.4, 42.3, 41.2, 37.5, 37.4, 36.7, 32.1, 32.1, 31.8, 28.4, 24.9, 21.0, 19.6, 12.8, 9.6 (x2), 8.2.

#### 3.1.11. 5-(2-((17*R*)-3β-((*tert*-Butyldimethylsilyl)oxy)-androst-5-en-17-yl)acetyl)-4-cyclopropyloxazol-2(3*H*)-one (**28**)

A mixture of **26c** (20 mg, 31 μmol), THF (500 μL), water (200 μL), and NaOH (6.4 mg, 160 μmol) was heated at 70 °C for 36 h. Then, it was neutralized with saturated NH_4_Cl and extracted with EtOAc. The combined organic layers were washed with brine, dried over Na_2_SO_4_, and concentrated under reduced pressure. The crude product was purified by silica gel chromatography (PE:EtOAc, 90:10→70:30) to afford oxazolone **28** (12 mg, 66%) as a white solid. ^1^H NMR (500 MHz, CDCl_3_) δ 8.30 (s, 1H), 5.31 (dt, *J* = 5.0, 2.1 Hz, 1H), 3.48 (tt, *J* = 10.9, 4.7 Hz, 1H), 2.92 (dd, *J* = 14.7, 4.9 Hz, 1H), 2.68 (dd, *J* = 14.8, 9.1 Hz, 1H), 2.26 (ddd, *J* = 13.6, 11.0, 2.7 Hz, 1H), 2.18 (tt, *J* = 7.9, 4.7 Hz, 2H), 2.03–1.94 (m, 1H), 1.00 (s, 3H), 0.89 (s, 9H), 0.69 (s, 3H), 0.05 (s, 6H). ^13^C NMR (126 MHz, CDCl_3_) δ 189.2, 153.3, 149.5, 141.8, 123.2, 121.1, 72.7, 55.8, 50.5, 49.3, 43.0, 42.4, 37.6, 37.4, 36.8, 32.2, 32.1, 32.0, 28.2, 27.7, 26.1 (x3), 24.6, 20.9, 19.6, 18.9, 18.4, 12.5, 11.74, 11.70, −4.4 (x2).

#### 3.1.12. General Procedure for the Synthesis of Alcohols (**24a**–**i**)

A 1M solution of silyl ethers **23a**–**i** (1 eq.) and TBAF (1.2 eq.) in THF was kept at room temperature for 24 h. On completion of the reaction, the mixture was diluted with saturated NH_4_Cl and extracted with EtOAc. The combined organic layers were washed with water, brine, dried over Na_2_SO_4_ and concentrated under reduced pressure. The residue was purified by silica gel chromatography (PE:EtOAc) to give alcohols **24a**–**i**.

##### (17*R*)-(Isoxazol-5-ylmethyl)-androst-5-en-3β-ol (**24a**)

The title compound **24a** (70 mg) was prepared as a white solid in 73% yield from silyl ether **23a**. ^1^H NMR (500 MHz, CDCl_3_) δ 8.12 (d, *J* = 1.7 Hz, 1H), 5.96 (d, *J* = 1.7 Hz, 1H), 5.35 (dt, *J* = 5.5, 2.0 Hz, 1H), 3.52 (tt, *J* = 11.2, 4.6 Hz, 1H), 2.85 (dd, *J* = 15.0, 5.1 Hz, 1H), 2.59 (dd, *J* = 15.0, 9.7 Hz, 1H), 2.30 (ddd, *J* = 13.1, 5.1, 2.1 Hz, 1H), 2.23 (ddq, *J* = 13.4, 11.0, 2.7 Hz, 1H), 1.99 (dtd, *J* = 16.9, 4.9, 2.6 Hz, 1H), 1.01 (s, 3H), 0.69 (s, 3H). ^13^C NMR (126 MHz, CDCl_3_) δ 173.0, 150.3, 141.0, 121.7, 100.3, 71.9, 55.9, 50.5, 49.1, 42.4, 42.3, 37.5, 37.4, 36.7, 32.1, 32.0, 31.8, 28.5, 27.6, 24.7, 20.9, 19.6, 12.4.

##### (17*R*)-17-((3-isopropylisoxazol-5-yl)methyl)-androst-5-en-3β-ol (**24b**)

The title compound **24b** (31 mg) was prepared as a white solid in 85% yield from silyl ether **23b**. ^1^H NMR (500 MHz, CDCl_3_) δ 5.81 (s, 1H), 5.34 (dt, *J* = 4.9, 2.0 Hz, 1H), 3.51 (tt, *J* = 11.2, 4.5 Hz, 1H), 3.00 (hept, *J* = 6.9 Hz, 1H), 2.78 (dd, *J* = 15.1, 5.0 Hz, 1H), 2.50 (dd, *J* = 15.0, 10.0 Hz, 1H), 2.35–2.18 (m, 2H), 1.98 (dtd, *J* = 17.0, 5.0, 2.6 Hz, 1H), 1.25 (d, *J* = 7.0 Hz, 6H), 1.01 (s, 3H), 0.67 (s, 3H). ^13^C NMR (126 MHz, CDCl_3_) δ 173.2, 169.4, 141.0, 121.6, 99.1, 71.8, 55.9, 50.5, 49.0, 42.4, 42.3, 37.4 (x2), 36.7, 32.1, 32.0, 31.8, 28.6, 27.7, 26.6, 24.7, 21.9 (x2), 20.9, 19.5, 12.4. HRMS (ESI): *m*/*z* calcd for C_26_H_40_NO_2_ [M+H]^+^: 398.3054, found 398.3064.

##### (17*R*)-17-((3-Cyclopropylisoxazol-5-yl)methyl)-androst-5-en-3β-ol (**24c**)

The title compound **24c** (27 mg) was prepared as a white solid in 70% yield from silyl ether **23c**. ^1^H NMR (500 MHz, CDCl_3_) δ 5.60 (s, 1H), 5.34 (dt, *J* = 5.3, 2.0 Hz, 1H), 3.51 (tt, *J* = 11.2, 4.6 Hz, 1H), 2.75 (dd, *J* = 15.0, 5.0 Hz, 1H), 2.47 (dd, *J* = 15.0, 10.0 Hz, 1H), 2.29 (ddd, *J* = 13.1, 5.1, 2.2 Hz, 1H), 2.22 (ddq, *J* = 13.4, 11.1, 2.6 Hz, 1H), 1.01 (s, 3H), 1.00–0.96 (m, 2H), 0.76 (dt, *J* = 6.8, 4.6 Hz, 2H), 0.66 (s, 3H). ^13^C NMR (126 MHz, CDCl_3_) δ 173.3, 166.4, 141.0, 121.6, 98.4, 71.8, 55.9, 50.5, 49.0, 42.4, 42.3, 37.4 (x2), 36.7, 32.1, 32.0, 31.8, 28.6, 27.7, 24.7, 20.9, 19.6, 12.4, 8.0 (x2), 7.5. HRMS (ESI): *m*/*z* calcd for C_26_H_38_NO_2_ [M+H]^+^: 396.2897, found 396.2908.

##### 17β-((3-Butylisoxazol-5-yl)methyl)-androst-5-en-3β-ol (**24d**)

The title compound **24d** (53 mg) was prepared as a white solid in 70% yield from silyl ether **23d**. ^1^H NMR (500 MHz, CDCl_3_) δ 5.79 (s, 1H), 5.34 (dt, *J* = 5.4, 2.0 Hz, 1H), 3.52 (tt, *J* = 11.2, 4.6 Hz, 1H), 2.78 (dd, *J* = 15.0, 5.1 Hz, 1H), 2.65–2.55 (m, 2H), 2.51 (dd, *J* = 15.1, 9.9 Hz, 1H), 2.32–2.19 (m, 2H), 1.99 (dtd, *J* = 16.9, 5.0, 2.5 Hz, 1H), 1.01 (s, 3H), 0.92 (t, *J* = 7.4 Hz, 3H), 0.67 (s, 3H). ^13^C NMR (126 MHz, CDCl_3_) δ 173.2, 164.2, 141.0, 121.6, 100.7, 71.9, 55.9, 50.5, 49.0, 42.4, 42.3, 37.5, 37.4, 36.7, 32.1, 32.0, 31.8, 30.6, 28.6, 27.7, 25.9, 24.7, 22.4, 20.9, 19.6, 13.9, 12.4. 

##### (17*R*)-17-((3-(Pyridin-3-yl)isoxazol-5-yl)methyl)-androst-5-en-3β-ol (**24f**)

The title compound **24f** (15 mg) was prepared as a white solid in 64% yield from silyl ether **23f**. ^1^H NMR (500 MHz, CDCl_3_) δ 9.01 (s, 1H), 8.69 (s, 1H), 8.14 (d, *J* = 7.9 Hz, 1H), 7.40 (s, 1H), 6.33 (s, 1H), 5.35 (dd, *J* = 5.0, 2.4 Hz, 1H), 3.52 (tt, *J* = 11.2, 4.4 Hz, 1H), 2.90 (dd, *J* = 15.1, 5.1 Hz, 1H), 2.63 (dd, *J* = 15.1, 9.8 Hz, 1H), 2.34–2.27 (m, 1H), 2.23 (ddd, *J* = 13.5, 11.0, 2.8 Hz, 1H), 1.02 (s, 3H), 0.72 (s, 3H). ^13^C NMR (126 MHz, CDCl_3_) δ 175.0, 159.9, 150.9, 148.0, 141.0, 134.1, 121.6, 99.0, 77.4, 77.2, 76.9, 71.9, 55.9, 50.5, 49.1, 42.4, 42.4, 37.5, 37.4, 36.7, 32.1, 32.0, 31.8, 28.6, 27.9, 24.7, 21.0, 19.6, 12.5. HRMS (ESI): *m*/*z* calcd for C_28_H_37_N_2_O_2_ [M+H]^+^: 433.2850, found 433.2862.

##### 17β-((3-(2-Fluorophenyl)isoxazol-5-yl)methyl)-androst-5-en-3β-ol (**24g**)

The title compound **24g** (9 mg) was prepared as a white solid in 82% yield from silyl ether **23g**. ^1^H NMR (500 MHz, CDCl_3_) δ 7.97 (td, *J* = 7.6, 1.9 Hz, 1H), 7.45–7.36 (m, 1H), 7.22 (td, *J* = 7.6, 1.2 Hz, 1H), 7.19–7.13 (m, 1H), 6.43 (d, *J* = 3.7 Hz, 1H), 5.36 (dd, *J* = 4.9, 2.3 Hz, 1H), 3.52 (dq, *J* = 11.3, 5.7, 5.0 Hz, 1H), 2.89 (dd, *J* = 15.1, 5.1 Hz, 1H), 2.63 (dd, *J* = 15.1, 9.7 Hz, 1H), 2.30 (ddd, *J* = 13.1, 5.1, 2.1 Hz, 1H), 2.23 (ddd, *J* = 13.6, 10.6, 2.6 Hz, 1H), 2.00 (dtd, *J* = 16.9, 4.9, 2.5 Hz, 1H), 1.02 (s, 3H), 0.72 (s, 3H). ^13^C NMR (126 MHz, CDCl_3_) δ 174.1, 160.4 (d, *J* = 251.4 Hz), 157.8, 141.0, 131.5 (d, *J* = 8.6 Hz), 129.2 (d, *J* = 3.1 Hz), 124.7 (d, *J* = 3.5 Hz), 121.7, 117.6 (d, *J* = 12.1 Hz), 116.4 (d, *J* = 21.9 Hz), 101.8 (d, *J* = 9.1 Hz), 71.9, 55.9, 50.5, 49.1, 42.4, 42.3, 37.5, 37.4, 36.7, 32.1, 32.0, 31.8, 28.6, 27.8, 24.7, 21.0, 19.6, 12.5.

#### 3.1.13. (17*R*)-17-((3-(2-Hydroxypropan-2-yl)isoxazol-5-yl)methyl)-androst-5-en-3β-ol (**24j**)

A mixture of silyl ether **23h** (270 mg, 0.44 mmol), THF (1 mL), MeCN (4 mL), and 40% aq. HF (100 μL, 2 mmol) in a Teflon vial was stirred at room temperature for 16 h. Then, it was neutralized with saturated NaHCO_3_ and extracted with EtOAc. The combined organic layers were washed with water, brine, dried over Na_2_SO_4_, and concentrated under reduced pressure. The residue was purified by silica gel chromatography (CHCl_3_: MeOH, 100:0→80:20) to give diol **24j** (128 mg, 70%) as a white solid. ^1^H NMR (500 MHz, CDCl_3_) δ 5.97 (s, 1H), 5.35 (dd, *J* = 4.8, 2.3 Hz, 1H), 3.52 (tt, *J* = 11.1, 4.6 Hz, 1H), 2.80 (dd, *J* = 15.1, 4.9 Hz, 1H), 2.53 (dd, *J* = 15.1, 10.1 Hz, 1H), 2.30 (ddd, *J* = 13.1, 5.1, 2.0 Hz, 1H), 2.23 (ddd, *J* = 13.2, 10.7, 2.6 Hz, 1H), 1.58 (s, 6H), 1.01 (s, 3H), 0.68 (s, 3H). ^13^C NMR (126 MHz, CDCl_3_) δ 174.0, 170.5, 141.0, 121.6, 98.6, 71.9, 69.4, 55.9, 50.5, 49.1, 42.4, 42.3, 37.5, 37.4, 36.7, 32.1, 32.0, 31.8, 30.0 (x2), 28.6, 27.8, 24.7, 21.0, 19.6, 12.4. HRMS (ESI): *m*/*z* calcd for C_26_H_40_NO_3_ [M+H]^+^: 414.3003, found 414.3014.

#### 3.1.14. (17*R*)-17-((3-Phenylisoxazol-5-yl)methyl)-androst-5-en-3β-ol (**24e**)

The title compound **24e** (30 mg) was obtained from **23e** in a similar manner for the preparation of **24j** as a white solid in 88% yield. ^1^H NMR (500 MHz, CDCl_3_) δ 7.83–7.73 (m, 2H), 7.44 (d, *J* = 6.8 Hz, 3H), 6.28 (s, 1H), 5.40–5.30 (m, 1H), 3.52 (ddd, *J* = 18.6, 12.0, 5.9 Hz, 1H), 2.88 (dd, *J* = 15.1, 5.1 Hz, 1H), 2.61 (dd, *J* = 15.1, 9.7 Hz, 1H), 2.36–2.18 (m, 2H), 1.02 (s, 3H), 0.71 (s, 3H). ^13^C NMR (126 MHz, CDCl_3_) δ 174.2, 162.4, 141.0, 129.9, 129.6, 129.0 (x2), 126.9 (x2), 121.7, 99.2, 71.9, 55.9, 50.5, 49.1, 42.4, 42.3, 37.5, 37.4, 36.7, 32.1, 32.0, 31.8, 28.6, 27.8, 24.7, 20.9, 19.6, 12.5. HRMS (ESI): *m*/*z* calcd for C_29_H_38_NO_2_ [M+H]^+^: 432.2897, found 432.2910.

#### 3.1.15. (17*R*)-17-((3-(Hydroxymethyl)isoxazol-5-yl)methyl)-androst-5-en-3β-ol (**32**)

Dowex 50 W (H^+^-form) ion-exchanger (2 mg) was added to a solution of **24i** (40 mg, 6.8 μmol) in THF (340 μL) and MeOH (800 μL). The suspension was stirred at room temperature for 48 h. Solids were filtered off and washed with EtOAc. The filtrate was evaporated under reduced pressure, and the crude material was purified by silica gel chromatography (CHCl_3_:MeOH, 100:0→80:20) to give diol **32** (23 mg, 87%) as a white solid. ^1^H NMR (500 MHz, CDCl_3_) δ 6.02 (s, 1H), 5.35 (dt, *J* = 4.7, 2.0 Hz, 1H), 4.71 (s, 2H), 3.51 (tt, *J* = 11.2, 4.6 Hz, 1H), 2.82 (dd, *J* = 15.1, 5.1 Hz, 1H), 2.55 (dd, *J* = 15.1, 9.9 Hz, 1H), 2.29 (ddd, *J* = 13.0, 5.1, 2.1 Hz, 1H), 2.26–2.18 (m, 4H), 1.99 (dtd, *J* = 16.8, 5.0, 2.6 Hz, 1H), 1.01 (s, 3H), 0.68 (s, 3H). ^13^C NMR (126 MHz, CDCl_3_) δ 174.2, 163.5, 141.0, 121.6, 99.9, 71.9, 57.3, 55.9, 50.5, 49.1, 42.4, 42.3, 37.5, 37.4, 36.7, 32.1, 32.0, 31.8, 28.6, 27.8, 24.7, 20.9, 19.6, 12.4. HRMS (ESI): *m*/*z* calcd for C_24_H_36_NO_3_ [M+H]^+^: 386.2690, found 386.2690.

#### 3.1.16. 5-(((17*R*)-3β-((*tert*-Butyldimethylsilyl)oxy)-androst-5-en-17-yl)methyl)isoxazole-3-yl)methanol (**33**)

A mixture of magnesium bromide diethyl etherate (prepared from Mg (63.4 mg, 2.61 mmol) by dropwise addition of 1,2-dibromoethane (222 μL, 2.56 mmol) in anhydrous Et_2_O (3 mL) and subsequent reflux until gas evolution subsides) and ether **24i** (440 mg, 0.75 mmol) was vigorously stirred at room temperature for 16 h. The mixture was quenched with saturated NH_4_Cl and extracted with EtOAc. Organic extracts were washed with brine, dried over Na_2_SO_4_, and rotoevaporated. The residue was purified by silica gel chromatography (PE:EtOAc, 95:5→70:30) to afford alcohol **33** as a white solid (323 mg, 85%), in addition to the unreacted starting material **24i** (43 mg, 9.8%). ^1^H NMR (500 MHz, CDCl_3_) δ 6.01 (s, 1H), 5.31 (dt, *J* = 5.1, 2.1 Hz, 1H), 4.69 (s, 2H), 3.47 (tt, *J* = 11.0, 4.7 Hz, 1H), 2.81 (dd, *J* = 15.1, 5.0 Hz, 1H), 2.54 (dd, *J* = 15.1, 9.9 Hz, 1H), 2.48–2.35 (m, 1H), 2.26 (tq, *J* = 11.1, 2.7 Hz, 1H), 2.16 (ddd, *J* = 13.3, 5.1, 2.3 Hz, 1H), 1.98 (dtd, *J* = 16.8, 5.0, 2.6 Hz, 1H), 1.00 (s, 3H), 0.88 (s, 9H), 0.67 (s, 3H), 0.05 (s, 6H). ^13^C NMR (126 MHz, CDCl_3_) δ 174.2, 163.6, 141.7, 121.1, 99.9, 72.7, 57.2, 55.9, 50.5, 49.0, 42.9, 42.3, 37.5, 37.5, 36.8, 32.2, 32.1, 32.0, 28.5, 27.7, 26.1 (x3), 24.7, 20.9, 19.6, 18.4, 12.4, −4.5 (x2).

#### 3.1.17. 5-(((17*R*)-3β-((*tert*-Butyldimethylsilyl)oxy)-androst-5-en-17-yl)methyl)-3-(chloromethyl)isoxazole (**37**)

To a solution of alcohol **33** (212 mg, 0.42 mmol) in pyridine (0.5 mL), TsCl (122, 0.64 mmol) was added at 0 °C. The cooling bath was removed, and the reaction mixture was kept at room temperature for 16 h. On completion of the reaction, the mixture was diluted with water and extracted with EtOAc. The combined organics were washed with brine, dried over Na_2_SO_4_, and rotoevaporated. The residue was purified by silica gel chromatography (PE:EtOAc, 95:5→85:15) to give chloride **37** (43 mg, 20%) as a white solid. ^1^H NMR (500 MHz, CDCl_3_) δ 6.08 (s, 1H), 5.31 (dt, *J* = 5.5, 2.0 Hz, 1H), 4.55 (s, 2H), 3.48 (tt, *J* = 11.0, 4.7 Hz, 1H), 2.82 (dd, *J* = 15.2, 5.0 Hz, 1H), 2.55 (dd, *J* = 15.1, 9.9 Hz, 1H), 2.26 (ddd, *J* = 13.7, 10.9, 2.7 Hz, 1H), 2.17 (ddd, *J* = 13.4, 5.0, 2.3 Hz, 1H), 1.00 (s, 3H), 0.89 (s, 9H), 0.68 (s, 3H), 0.05 (s, 6H). ^13^C NMR (126 MHz, CDCl_3_) δ 174.8, 160.8, 141.8, 129.6, 121.1, 100.8, 72.7, 55.9, 50.6, 49.0, 43.0, 42.3, 37.6, 37.5, 36.8, 36.0, 32.2, 32.1, 32.1, 28.6, 27.8, 26.1 (x3), 24.7, 20.9, 19.6, 18.4, 12.4, −4.4 (x2). HRMS (ESI): *m*/*z* calcd for C_30_H_49_ClNO_2_Si [M+H]^+^: 518.3216, found 518.3226.

#### 3.1.18. 5-((3β-((*tert*-Butyldimethylsilyl)oxy)-androst-5-en-17-yl)methyl)isoxazol-3-yl)methyl methanesulfonate (**34**)

To a solution of alcohol **33** (250 mg, 0.5 mmol) in DCM (2 mL) and Et_3_N (120 μL, 0.86 mmol), MsCl (55 μL, 0.71 mmol) was added at −15 °C. The mixture was stirred at this temperature for 15 min, then quenched with saturated NaHCO_3_. The organic layer was separated, the water phase was extracted with CH_2_Cl_2_. The combined organics were washed with brine, dried over Na_2_SO_4_, and evaporated under reduced pressure. The residue was purified by silica gel chromatography (PE:EtOAc, 95:5→80:20) to give mesylate **34** (266 mg, 92%) as a white solid. ^1^H NMR (500 MHz, CDCl_3_) δ 6.13 (s, 1H), 5.31 (dt, *J* = 5.5, 2.0 Hz, 1H), 5.26 (s, 2H), 3.47 (tt, *J* = 10.7, 4.7 Hz, 1H), 3.05 (s, 3H), 2.84 (dd, *J* = 15.2, 5.0 Hz, 1H), 2.57 (dd, *J* = 15.2, 9.9 Hz, 1H), 2.26 (ddd, *J* = 13.7, 10.9, 2.7 Hz, 1H), 2.17 (ddd, *J* = 13.3, 5.0, 2.3 Hz, 1H), 1.99 (dtd, *J* = 16.8, 5.0, 2.5 Hz, 1H), 1.00 (s, 3H), 0.88 (s, 9H), 0.68 (s, 3H), 0.05 (s, 6H). ^13^C NMR (126 MHz, CDCl_3_) δ 175.4, 158.1, 141.8, 121.1, 100.8, 72.7, 62.3, 55.9, 50.5, 49.0, 43.0, 42.3, 38.3, 37.6, 37.5, 36.8, 32.2, 32.1, 32.0, 28.5, 27.8, 26.1 (x3), 24.7, 20.9, 19.6, 18.4, 12.4, −4.4 (x2). HRMS (ESI): *m*/*z* calcd for C_31_H_52_NO_5_SSi [M+H]^+^: 578.3330, found 578.3333.

#### 3.1.19. 3-(Azidomethyl)-5-(((17*R*)-3β-((*tert*-butyldimethylsilyl)oxy)-androst-5-en-17-yl)methyl)isoxazole (**35**)

A mixture of mesylate **34** (260 mg, 0.45 mmol), NaN_3_ (88 mg, 1.35 mmol), and DMF (2 mL) was stirred at 70 °C for 3 h. On completion of the reaction, the mixture was diluted with water and extracted with EtOAc. The organic layers were washed with water, brine, dried over Na_2_SO_4_, and concentrated under reduced pressure. The resulting product was purified by silica gel chromatography (PE:EtOAc, 100:0→85:15) to afford azide **35** (220 mg, 93%) as a white solid. ^1^H NMR (500 MHz, CDCl3) δ 6.02 (s, 1H), 5.31 (dt, *J* = 5.5, 2.0 Hz, 1H), 4.37 (s, 2H), 3.47 (tt, *J* = 11.1, 4.7 Hz, 1H), 2.84 (dd, *J* = 15.1, 5.1 Hz, 1H), 2.57 (dd, *J* = 15.1, 9.8 Hz, 1H), 2.31–2.22 (m, 1H), 2.17 (ddd, *J* = 13.4, 5.0, 2.3 Hz, 1H), 1.99 (dtd, *J* = 16.8, 4.9, 2.6 Hz, 1H), 1.00 (s, 3H), 0.88 (s, 9H), 0.68 (s, 3H), 0.05 (s, 6H). ^13^C NMR (126 MHz, CDCl_3_) δ 175.0, 159.1, 141.8, 121.1, 100.3, 72.7, 55.9, 50.5, 49.0, 45.9, 43.0, 42.3, 37.6, 37.5, 36.8, 32.2, 32.1, 32.1, 28.5, 27.8, 26.1 (x3), 24.7, 20.9, 19.6, 18.4, 12.4, −4.4 (x2).

#### 3.1.20. (17*R*)-17-((3-(Azidomethyl)isoxazol-5-yl)methyl)-androst-5-en-3β-ol (**36**)

The title compound **36** (25 mg) was obtained from **35** in a similar manner for the preparation of **24j** as a white solid in 81% yield. ^1^H NMR (500 MHz, CDCl_3_) δ 6.02 (s, 1H), 5.34 (dt, *J* = 4.8, 2.1 Hz, 1H), 4.36 (s, 2H), 3.52 (tt, *J* = 11.2, 4.5 Hz, 1H), 2.83 (dd, *J* = 15.1, 5.1 Hz, 1H), 2.57 (dd, *J* = 15.1, 9.8 Hz, 1H), 2.29 (ddd, *J* = 13.1, 5.2, 2.1 Hz, 1H), 2.23 (tq, *J* = 13.5, 2.6 Hz, 1H), 1.99 (dtd, *J* = 16.9, 5.0, 2.5 Hz, 1H), 1.01 (s, 3H), 0.69 (s, 3H). ^13^C NMR (126 MHz, CDCl_3_) δ 175.0, 159.1, 141.0, 121.6, 100.3, 71.8, 55.9, 50.4, 49.0, 45.9, 42.4, 42.3, 37.5, 37.4, 36.7, 32.1, 32.0, 31.8, 28.5, 27.8, 24.7, 20.9, 19.5, 12.4.

#### 3.1.21. (17*R*)-17-((3-(Chloromethyl)isoxazol-5-yl)methyl)-androst-5-en-3β-ol (**38**)

The title compound **38** (26 mg) was obtained from **37** in a similar manner for the preparation of **24a** as a white solid in 85% yield. ^1^H NMR (500 MHz, CDCl_3_) δ 6.08 (s, 1H), 5.39–5.32 (m, 1H), 4.54 (s, 1H), 3.52 (tt, *J* = 11.2, 4.6 Hz, 1H), 2.82 (dd, *J* = 15.1, 5.0 Hz, 1H), 2.55 (dd, *J* = 15.1, 10.2 Hz, 1H), 2.30 (ddd, *J* = 13.1, 5.1, 2.1 Hz, 1H), 2.26–2.19 (m, 1H), 1.01 (s, 3H), 0.68 (s, 3H). ^13^C NMR (126 MHz, CDCl_3_) δ 174.8, 160.8, 141.0, 121.6, 100.8, 71.9, 55.9, 50.5, 49.0, 42.4, 42.3, 37.4, 37.4, 36.7, 36.0, 32.1, 32.0, 31.8, 28.5, 27.8, 24.7, 20.9, 19.6, 12.4. HRMS (ESI): *m*/*z* calcd for C_24_H_35_ClNO_2_ [M+H]^+^: 404.2351, found 404.2358.

#### 3.1.22. 1-((17*R*)-3β-((*tert*-Butyldimethylsilyl)oxy)-androst-5-en-17-yl)oct-3-yn-2-one oxime (**39**)

A mixture of ynone **20d** (100 mg, 0.20 mmol), NH_2_OH∙HCl (42 mg, 0.60 mmol), NaHCO_3_ (50 mg, 060 mmol), and methanol (1.5 mL) was stirred at 60 °C for 40 min, then evaporated to dryness at reduced pressure and purified by silica gel chromatography (PE:EtOAc, 100:0→80:20), affording oxime **39** as an oil. ^1^H NMR (500 MHz, CDCl_3_) δ 5.31 (dt, *J* = 5.6, 2.0 Hz, 1H), 3.47 (td, *J* = 11.0, 5.3 Hz, 1H), 1.01, 1.00 (s, 3H), 0.89 (s, 9H), 0.66 (s, 3H), 0.63, 0.62 (s, 3H), 0.06 (s, 6H). ^13^C NMR (126 MHz, CDCl_3_) δ 162.3, 148.4, 144.0, 141.8, 121.2, 103.6, 92.7, 72.8, 56.1, 55.9, 55.9, 50.7, 48.1, 47.6, 47.5, 47.5, 43.0, 42.6, 42.4, 42.3, 37.7, 37.6, 37.5, 37.5, 36.9, 35.6, 34.7, 34.0, 32.2, 32.2, 32.1, 30.5, 30.4, 29.7, 28.7, 28.5, 28.3, 28.1, 28.0, 27.3, 26.1, 24.8, 24.8, 23.1, 22.6, 22.1, 21.0, 19.6, 19.5, 19.1, 18.4, 14.0, 13.7, 12.5, 12.4, 12.4, 12.3, −4.4.

#### 3.1.23. 5-Butyl-3-(((17*R*)-3β-((*tert*-butyldimethylsilyl)oxy)-androst-5-en-17-yl)methyl)isoxazole (**40d**)

A mixture of oxime **39** (93 mg, 0.18 mmol), AuCl_3_ (1.5 mg, 0.005 mmol), and DCM (0.7 mL) was stirred at room temperature for 30 min. The solvent was evaporated, and the residue was purified by silica gel chromatography (PE:EtOAc, 100:0→85:15) to give isoxazole **40d** (43 mg, 46%) as an oil. ^1^H NMR (500 MHz, CDCl_3_) δ 5.79 (d, *J* = 0.9 Hz, 1H), 5.31 (dt, *J* = 5.6, 1.9 Hz, 1H), 3.52–3.43 (m, 1H), 2.75–2.65 (m, 2H), 2.48–2.37 (m, 1H), 1.01 (s, 3H), 0.93 (t, *J* = 7.4 Hz, 3H), 0.89 (s, 9H), 0.69 (s, 3H), 0.05 (s, 6H). ^13^C NMR (126 MHz, CDCl_3_) δ 173.3, 163.9, 141.8, 121.2, 100.7, 72.8, 56.1, 50.7, 49.8, 43.0, 42.4, 37.7, 37.6, 36.8, 32.2, 32.1, 29.7, 28.6, 26.9, 26.6, 26.1 (x3), 24.7, 22.3, 21.0, 19.6, 18.4, 13.8, 12.4, −4.4 (x2).

#### 3.1.24. General Procedure for the Synthesis of Isoxazoles **40a**,**e**–**i**

A 0.1M methanol solution of ynones **20a**,**e**–**i** (1 eq) was added to NH_2_OH∙HCl (3 eq) and NaHCO_3_ (3 eq). The resulting mixture was stirred at 60 °C for 16 h, then the solvent was evaporated at reduced pressure and the residue was purified by silica gel chromatography (PE:EtOAc) to afford isoxazoles **40a**,**e**–**i**.

##### 3-(((17*R*)-3β-((*tert*-Butyldimethylsilyl)oxy)-androst-5-en-17-yl)methyl)isoxazole (**40a**)

The title compound **40a** (47 mg) was prepared as a white solid in 61% yield from ynone **20a**. ^1^H NMR (500 MHz, CDCl_3_) δ 8.28 (s, 1H), 6.18 (s, 1H), 5.31 (d, *J* = 5.0 Hz, 1H), 3.52–3.43 (m, 1H), 2.81 (dd, *J* = 14.6, 4.9 Hz, 1H), 2.54–2.46 (m, 1H), 2.27 (t, *J* = 12.2 Hz, 1H), 2.21–2.13 (m, 1H), 1.99 (d, *J* = 16.8 Hz, 1H), 1.00 (s, 3H), 0.88 (s, 9H), 0.70 (s, 3H), 0.05 (s, 6H). ^13^C NMR (126 MHz, CDCl_3_) δ 162.9, 158.0, 141.8, 121.2, 104.4, 72.8, 56.1, 50.6, 49.8, 43.0, 42.4, 37.7, 37.6, 36.8, 32.2, 32.1, 29.9, 28.6, 26.7, 26.1 (x3), 24.7, 21.0, 19.6, 18.4, 12.5, −4.4 (x2).

##### 3-(((17*R*)-3β-((*tert*-Butyldimethylsilyl)oxy)-androst-5-en-17-yl)methyl)-5-phenylisoxazole (**40e**)

The title compound **40e** (140 mg) was prepared as a white solid in 82% yield from ynone **20e**. ^1^H NMR (500 MHz, CDCl_3_) δ 7.80–7.73 (m, 2H), 7.49–7.38 (m, 3H), 6.37 (s, 1H), 5.32 (dd, *J* = 4.8, 2.7 Hz, 1H), 3.48 (tt, *J* = 11.0, 4.7 Hz, 1H), 2.81 (dd, *J* = 14.3, 4.9 Hz, 1H), 2.51 (dd, *J* = 14.3, 10.1 Hz, 1H), 2.26 (ddt, *J* = 13.7, 5.5, 2.8 Hz, 1H), 2.17 (ddd, *J* = 13.3, 5.0, 2.2 Hz, 1H), 2.04–1.95 (m, 1H), 1.01 (s, 3H), 0.89 (s, 9H), 0.73 (s, 3H), 0.06 (s, 6H). ^13^C NMR (126 MHz, CDCl_3_) δ 169.5, 164.6, 141.8, 130.1, 129.1 (x2), 127.9, 125.9 (x2), 121.2, 99.6, 72.8, 56.1, 50.7, 49.8, 43.0, 42.5, 37.8, 37.6, 36.8, 32.2, 32.1 (x2), 28.6, 27.0, 26.1 (x3), 24.8, 21.0, 19.6, 18.4, 12.5, −4.4 (x2). HRMS (ESI): *m*/*z* calcd for C_35_H_52_NO_2_Si [M+H]^+^: 546.3762, found 546.3772.

##### 3-(((17*R*)-3β-((*tert*-Butyldimethylsilyl)oxy)-androst-5-en-17-yl)methyl)-5-(pyridin-3-yl)isoxazole (**40f**)

The title compound **40f** (40 mg) was prepared as a white solid in 40% yield from ynone **20f**. ^1^H NMR (500 MHz, CDCl_3_) δ 8.99 (d, *J* = 2.3 Hz, 1H), 8.65 (dd, *J* = 4.9, 1.7 Hz, 1H), 8.07 (dt, *J* = 8.0, 2.0 Hz, 1H), 7.40 (ddd, *J* = 8.0, 4.8, 0.9 Hz, 1H), 6.47 (s, 1H), 5.31 (dt, *J* = 5.7, 1.9 Hz, 1H), 3.47 (tt, *J* = 11.0, 4.7 Hz, 1H), 2.83 (dd, *J* = 14.3, 4.9 Hz, 1H), 2.53 (dd, *J* = 14.3, 10.1 Hz, 1H), 2.26 (ddd, *J* = 13.7, 10.9, 2.7 Hz, 1H), 2.17 (ddd, *J* = 13.3, 5.0, 2.3 Hz, 1H), 1.01 (s, 3H), 0.88 (s, 9H), 0.72 (s, 3H), 0.05 (s, 6H). ^13^C NMR (126 MHz, CDCl_3_) δ 166.6, 164.8, 150.8, 147.1, 141.8, 133.0, 124.1, 123.9, 121.1, 100.6, 72.7, 56.1, 50.6, 49.8, 43.0, 42.5, 37.7, 37.6, 36.8, 32.2, 32.1, 28.6, 27.0, 26.1 (x2), 24.7, 20.9, 19.6, 18.4, 12.5, −4.4 (x2).

##### 3-(((17*R*)-3β-((*tert*-Butyldimethylsilyl)oxy)-androst-5-en-17-yl)methyl)-5-(2-fluorophenyl)isoxazole (**40g**)

The title compound **40g** (97 mg) was prepared as a white solid in 69% yield from ynone **20g**. ^1^H NMR (500 MHz, CDCl_3_) δ 7.95 (td, *J* = 7.6, 1.7 Hz, 1H), 7.40 (tdd, *J* = 7.4, 5.0, 1.8 Hz, 1H), 7.29–7.22 (m, 1H), 7.18 (dd, *J* = 11.0, 8.2 Hz, 1H), 6.56 (d, *J* = 3.9 Hz, 1H), 5.32 (d, *J* = 5.2 Hz, 1H), 3.48 (tt, *J* = 10.9, 4.7 Hz, 1H), 2.83 (dd, *J* = 14.3, 4.9 Hz, 1H), 2.54 (dd, *J* = 14.3, 10.0 Hz, 1H), 2.27 (ddd, *J* = 13.9, 10.9, 2.8 Hz, 1H), 2.17 (ddd, *J* = 13.3, 5.1, 2.3 Hz, 1H), 1.02 (s, 3H), 0.89 (s, 9H), 0.73 (s, 3H), 0.06 (s, 6H). ^13^C NMR (126 MHz, CDCl_3_) δ 164.9, 163.2, 159.2 (d, *J* = 252.8 Hz), 141.8, 131.4 (d, *J* = 8.6 Hz), 127.8, 124.8, 124.8, 121.2, 116.3 (d, *J* = 21.3 Hz), 116.2, 103.8 (d, *J* = 10.9 Hz), 72.7, 56.1, 50.6, 49.7, 43.0, 42.5, 37.7, 37.6, 36.8, 32.2, 32.1, 28.6, 27.0, 26.1 (x3), 24.8, 21.0, 19.6, 18.4, 12.5, −4.4 (x2). HRMS (ESI): *m*/*z* calcd for C_35_H_51_FNO_2_Si [M+H]^+^: 564.3668, found 564.3678.

##### 3-(((17*R*)-3β-((*tert*-Butyldimethylsilyl)oxy)-androst-5-en-17-yl)methyl)-5-(2-((tetrahydro-2*H*-pyran-2-yl)oxy)propan-2-yl)isoxazole (**40h**)

The title compound **40h** (60 mg) was prepared as a white solid in 73% yield from ynone **20h**. ^1^H NMR (500 MHz, CDCl_3_) δ 5.99 (d, *J* = 1.0 Hz, 1H), 5.31 (dd, *J* = 4.7, 2.3 Hz, 1H), 4.59 (dt, *J* = 5.4, 2.5 Hz, 1H), 3.90 (dt, *J* = 11.3, 5.0 Hz, 1H), 3.47 (ddt, *J* = 15.7, 10.9, 4.7 Hz, 1H), 3.39 (dt, *J* = 11.1, 5.1 Hz, 1H), 2.74 (ddd, *J* = 14.3, 5.0, 2.2 Hz, 1H), 2.44 (dd, *J* = 14.3, 10.0 Hz, 1H), 2.26 (ddd, *J* = 13.7, 11.1, 2.9 Hz, 1H), 2.16 (ddd, *J* = 13.3, 5.0, 2.3 Hz, 1H), 1.64 (s, 3H), 1.58 (s, 3H), 1.00 (s, 3H), 0.88 (s, 9H), 0.69 (s, 3H), 0.05 (s, 6H). ^13^C NMR (126 MHz, CDCl_3_) δ 175.7, 175.6, 163.8, 163.7, 141.8, 121.1, 101.1, 101.0, 95.4, 95.3, 73.7, 73.6, 72.7, 63.3, 63.3, 56.1, 50.6, 49.7, 43.0, 42.4, 37.7, 37.6, 36.8, 32.2, 32.1, 32.0, 28.6, 28.6, 28.3, 28.2, 26.9, 26.2, 26.1, 25.4, 24.7, 21.0, 20.4, 20.4, 19.6, 18.4, 12.4, −4.4.

##### 3-(((17*R*)-3β-((*tert*-Butyldimethylsilyl)oxy)-androst-5-en-17-yl)methyl)-5-(((tetrahydro-2*H*-pyran-2-yl)oxy)methyl)isoxazole (**40i**)

The title compound **40i** (86 mg) was prepared as a white solid in 45% yield from ynone **20i**. ^1^H NMR (500 MHz, CDCl_3_) δ 6.09 (s, 1H), 5.31 (dt, *J* = 5.7, 1.9 Hz, 1H), 4.77–4.70 (m, 2H), 4.59 (d, *J* = 13.6 Hz, 1H), 3.86 (ddd, *J* = 11.6, 8.9, 2.9 Hz, 1H), 3.58–3.51 (m, 1H), 3.47 (tt, *J* = 10.9, 4.6 Hz, 1H), 2.75 (dd, *J* = 14.3, 4.7 Hz, 1H), 2.45 (dd, *J* = 14.3, 10.2 Hz, 1H), 2.26 (ddt, *J* = 13.8, 5.5, 3.0 Hz, 1H), 2.16 (ddd, *J* = 13.3, 5.0, 2.3 Hz, 1H), 1.00 (s, 3H), 0.88 (s, 9H), 0.69 (s, 3H), 0.05 (s, 6H). ^13^C NMR (126 MHz, CDCl_3_) δ 168.9, 163.9, 141.8, 121.2, 121.1, 103.0, 98.1, 72.7, 62.2, 59.8, 56.1, 50.6, 49.7, 43.0, 42.4, 37.7, 37.6, 36.8, 32.2, 32.1, 30.4, 28.6, 26.9, 26.1, 25.4, 24.7, 20.9, 19.6, 19.1, 18.4, 12.4, −4.4.

#### 3.1.25. (17*R*)-17-(Isoxazol-3-ylmethyl)-androst-5-en-3β-ol (**41a**)

The title compound **41a** (20 mg) was obtained from **40a** in a similar manner for the preparation of **24a** as a white solid in 87% yield. ^1^H NMR (500 MHz, CDCl_3_) δ 8.28 (d, *J* = 1.6 Hz, 1H), 6.18 (d, *J* = 1.7 Hz, 1H), 5.35 (dt, *J* = 5.5, 2.0 Hz, 1H), 3.52 (tt, *J* = 11.2, 4.6 Hz, 1H), 2.81 (dd, *J* = 14.3, 5.0 Hz, 1H), 2.50 (dd, *J* = 14.3, 10.0 Hz, 1H), 2.30 (ddd, *J* = 13.1, 5.1, 2.2 Hz, 1H), 2.23 (ddq, *J* = 13.3, 11.0, 2.5 Hz, 1H), 1.01 (s, 3H), 0.71 (s, 3H). ^13^C NMR (126 MHz, CDCl_3_) δ 162.9, 158.0, 141.0, 121.7, 104.4, 71.9, 56.0, 50.5, 49.8, 42.4, 37.7, 37.4, 36.7, 32.1, 32.1, 31.8, 28.5, 26.7, 24.7, 21.0, 19.6, 12.5. HRMS (ESI): *m*/*z* calcd for C_23_H_34_NO_2_ [M+H]^+^: 356.2584, found 356.2592.

#### 3.1.26. (17*R*)-17-((5-Butylisoxazol-3-yl)methyl)-androst-5-en-3β-ol (**41d**)

The title compound **41d** (63 mg) was obtained from **40d** in a similar manner for the preparation of **24a** as a white solid in 80% yield. ^1^H NMR (500 MHz, CDCl_3_) δ 5.79 (s, 1H), 5.34 (dt, *J* = 5.5, 2.0 Hz, 1H), 3.51 (tt, *J* = 11.1, 4.5 Hz, 1H), 2.76–2.65 (m, 3H), 2.40 (dd, *J* = 14.2, 10.2 Hz, 1H), 2.29 (ddd, *J* = 13.0, 5.1, 2.2 Hz, 1H), 2.22 (ddd, *J* = 13.4, 11.0, 2.8 Hz, 1H), 1.98 (dtd, *J* = 16.7, 4.7, 2.4 Hz, 1H), 1.01 (s, 3H), 0.92 (t, *J* = 7.4 Hz, 3H), 0.68 (s, 3H). ^13^C NMR (126 MHz, CDCl_3_) δ 173.3, 163.9, 141.0, 121.6, 100.7, 71.8, 56.0, 50.5, 49.7, 42.4, 42.4, 37.6, 37.4, 36.7, 32.1 (x2), 31.8, 29.7, 28.5, 26.9, 26.5, 24.7, 22.3, 20.9, 19.5, 13.8, 12.4. HRMS (ESI): *m*/*z* calcd for C_27_H_42_NO_2_ [M+H]^+^: 412.3210, found 412.3221.

#### 3.1.27. (17*R*)-17-((5-Phenylisoxazol-3-yl)methyl)-androst-5-en-3β-ol (**41e**)

The title compound **41e** (46 mg) was obtained from **40e** in a similar manner for the preparation of **24a** as a white solid in 85% yield. ^1^H NMR (500 MHz, CDCl_3_) δ 7.80–7.70 (m, 2H), 7.43 (dt, *J* = 11.4, 6.7 Hz, 3H), 6.37 (s, 1H), 5.35 (d, *J* = 5.1 Hz, 1H), 3.58–3.45 (m, 2H), 2.81 (dd, *J* = 14.2, 5.0 Hz, 1H), 2.51 (dd, *J* = 14.3, 10.0 Hz, 1H), 2.36–2.17 (m, 2H), 2.04–1.95 (m, 1H), 1.02 (s, 3H), 0.72 (s, 3H). ^13^C NMR (126 MHz, CDCl_3_) δ 169.5, 164.6, 141.0, 130.1, 129.0 (x2), 127.8, 125.9 (x2), 121.7, 99.5, 71.9, 56.0, 50.5, 49.8, 42.4 (x2), 37.7, 37.4, 36.7, 32.1, 32.1, 31.8, 28.6, 27.0, 24.7, 21.0, 19.6, 12.5. HRMS (ESI): *m*/*z* calcd for C_29_H_38_NO_2_ [M+H]^+^: 432.2897, found 432.2903.

#### 3.1.28. (17*R*)-17-((5-(Pyridin-3-yl)isoxazol-3-yl)methyl)-androst-5-en-3β-ol (**41f**)

The title compound **41f** (27 mg) was obtained from **40f** in a similar manner for the preparation of **24a** as a white solid in 62% yield. ^1^H NMR (500 MHz, CDCl_3_) δ 8.99 (s, 1H), 8.69–8.59 (m, 1H), 8.07 (dt, *J* = 8.0, 1.9 Hz, 1H), 7.41 (dd, *J* = 8.0, 4.8 Hz, 1H), 6.48 (s, 1H), 5.35 (dt, *J* = 5.5, 1.9 Hz, 1H), 3.52 (tt, *J* = 11.1, 4.5 Hz, 1H), 2.83 (dd, *J* = 14.3, 4.9 Hz, 1H), 2.54 (dd, *J* = 14.3, 10.1 Hz, 1H), 2.30 (ddd, *J* = 13.0, 5.1, 2.1 Hz, 1H), 2.24 (ddd, *J* = 13.2, 10.8, 2.6 Hz, 1H), 2.00 (dtd, *J* = 16.5, 4.7, 2.4 Hz, 1H), 1.02 (s, 3H), 0.73 (s, 3H). ^13^C NMR (126 MHz, CDCl_3_) δ 166.7, 164.8, 150.8, 147.1, 141.0, 133.0, 129.9, 123.9, 121.7, 100.6, 71.9, 56.1, 50.6, 49.8, 42.5, 42.4, 37.7, 37.4, 36.8, 32.1, 32.1, 31.8, 28.6, 27.0, 24.7, 21.0, 19.6, 12.5. HRMS (ESI): *m*/*z* calcd for C_28_H_37_N_2_O_2_ [M+H]^+^: 433.2850, found 433.2858.

#### 3.1.29. (17*R*)-17-((5-(2-Fluorophenyl)isoxazol-3-yl)methyl)-androst-5-en-3β-ol (**41g**)

The title compound **41g** (53 mg) was obtained from **40g** in a similar manner for the preparation of **24a** as a white solid in 90% yield. ^1^H NMR (500 MHz, CDCl_3_) δ 7.95 (td, *J* = 7.6, 1.8 Hz, 1H), 7.40 (tdd, *J* = 7.4, 5.0, 1.8 Hz, 1H), 7.26 (dq, *J* = 7.4, 3.9, 2.9 Hz, 2H), 7.18 (dd, *J* = 11.4, 8.0 Hz, 2H), 6.56 (d, *J* = 3.9 Hz, 1H), 5.35 (d, *J* = 5.1 Hz, 1H), 3.57–3.47 (m, 1H), 2.83 (dd, *J* = 14.3, 5.0 Hz, 1H), 2.54 (dd, *J* = 14.3, 10.0 Hz, 1H), 2.30 (ddd, *J* = 13.2, 5.2, 2.0 Hz, 1H), 2.24 (dt, *J* = 15.5, 7.7 Hz, 1H), 1.02 (s, 3H), 0.73 (s, 3H). ^13^C NMR (126 MHz, CDCl_3_) δ 164.9, 163.2, 159.2 (d, *J* = 252.8 Hz), 131.4 (d, *J* = 8.5 Hz), 131.4, 128.8 (d, *J* = 101.8 Hz), 124.8 (d, *J* = 3.5 Hz), 121.7, 116.3 (d, *J* = 21.7 Hz), 103.8 (d, *J* = 10.9 Hz), 71.9, 56.0, 50.5, 49.7, 42.4 (x2), 37.7, 37.4, 36.7, 32.1, 32.1, 31.8, 28.5, 27.0, 24.7, 21.0, 19.6, 12.5. HRMS (ESI): *m*/*z* calcd for C_29_H_37_FNO_2_ [M+H]^+^: 450.2803, found 450.2813.

#### 3.1.30. (17*R*)-17-((5-(2-Hydroxypropan-2-yl)isoxazol-3-yl)methyl)-androst-5-en-3β-ol (**41j**)

The title compound **41j** (182 mg) was obtained from **40h** in a similar manner for the preparation of **24j** as a white solid in 70% yield. ^1^H NMR (500 MHz, C_5_D_5_N) δ 6.42 (s, 1H), 5.43 (d, *J* = 4.9 Hz, 1H), 3.86 (td, *J* = 10.4, 5.7 Hz, 1H), 2.86 (dd, *J* = 14.2, 4.8 Hz, 1H), 2.68–2.59 (m, 2H), 2.55 (dd, *J* = 14.2, 9.6 Hz, 1H), 2.15–2.06 (m, 1H), 1.96 (dtd, *J* = 17.1, 5.2, 2.2 Hz, 1H), 1.83 (s, 6H), 1.06 (s, 3H), 0.66 (s, 3H). ^13^C NMR (126 MHz, C_5_D_5_N) δ 180.1, 164.2, 142.4, 121.5, 100.1, 71.6, 68.7, 56.4, 51.1, 50.2, 43.9, 42.7, 38.3, 38.0, 37.4, 33.0, 32.6, 32.5, 30.2, 30.1, 29.1, 27.5, 25.1, 21.5, 20.0, 12.7. HRMS (ESI): *m*/*z* calcd for C_26_H_40_NO_3_ [M+H]^+^: 414.3003, found 414.3013.

##### 3.1.31. (17*R*)-17-((5-(Hydroxymethyl)isoxazol-3-yl)methyl)-androst-5-en-3β-ol (**41k**)

The title compound **41k** (23 mg) was obtained from **40i** in a similar manner for the preparation of **24j** as a white solid in 90% yield. ^1^H NMR (500 MHz, C_5_D_5_N) δ 6.45 (s, 1H), 5.43 (d, *J* = 4.9 Hz, 1H), 5.04 (s, 2H), 3.87 (tt, *J* = 10.3, 5.4 Hz, 1H), 2.86 (dd, *J* = 14.2, 5.0 Hz, 1H), 2.65 (qd, *J* = 8.0, 2.3 Hz, 2H), 2.54 (dd, *J* = 14.2, 9.6 Hz, 1H), 2.15–2.07 (m, 1H), 1.06 (s, 3H), 0.66 (s, 3H). ^13^C NMR (126 MHz, C_5_D_5_N) δ 174.1, 164.5, 142.5, 121.6, 102.4, 71.7, 56.7, 56.5, 51.2, 50.3, 44.0, 42.8, 38.3, 38.2, 37.4, 33.1, 32.7, 32.6, 29.2, 27.5, 25.2, 21.5, 20.1, 12.8. HRMS (ESI): *m*/*z* calcd for C_24_H_36_NO_3_ [M+H]^+^: 386.2690, found 386.2694.

### 3.2. Biology

#### 3.2.1. CYP17A1 Inhibitory Assay

Recombinant human CYP17A1 was purified according to [38]. Recombinant rat NADPH-cytochrome P450 reductase (CPR) was purified according to [39]. Recombinant human cytochrome *b*_5_ was purified according to [40].

To determine ligand binding constants (Kd_app_ values) of the CYP17A1, spectrophotometric titration was performed using a Cary 5000 UV–Vis NIR dual-beam spectrophotometer (Agilent Technologies, Santa Clara, CA) in 1 cm quartz cuvettes. Stock solutions of the steroids were prepared at concentration 10 mM in DMSO. The titration was repeated at least three times, and Kd_app_ was calculated as described previously [41].

CYP17A1 activity was measured in the reconstituted system at 37 °C in 25 mM Hepes buffer (pH 7.2) according to early developed method [38]. Aliquots of concentrated recombinant proteins were mixed and pre-incubated for 5 min at RT. Progesterone or 17α-hydroxypregnenolone were added to the reaction mixture at the final concentration of 50 μM. Selected compounds were added to the reaction mixture at the final concentration of 50 μM. To measure the activity, the final concentrations of CYP17A1 and CPR were 1.0 and 2.0 μM, respectively. For analysis of 17,20-lyase activity, 1.0 μM cytochrome *b*_5_ was used. After 10 min of pre-incubation at 37 °C, the reaction was started by adding NADPH at the final concentration 0.25 mM. Aliquots (0.5 mL) were taken from the incubation mixture after 30 min of reaction. Steroids were extracted with 5 mL of methylene chloride. The organic layer was carefully removed and dried under argon flow. 100 μL of methanol was added to the pellet, and steroids were analyzed on a computerized HPLC system.

#### 3.2.2. Cultivation of Cell Lines

The reporter cell line ARE14 derived from 22Rv1 [36] was a kind gift from prof. Zdeněk Dvořák (Palacký University Olomouc, Czech Republic). LNCaP were purchased from ECACC, while LAPC-4 and DU145 cells were kindly gifted by prof. Jan Bouchal (Palacky University Olomouc and University Hospital, Olomouc, Czech Republic). ARE14, LNCaP, and DU145 were cultivated in RPMI-1640 medium, and LAPC-4 was cultivated in DMEM medium. All media were supplemented with 10% standard or charcoal-stripped fetal bovine serum (steroid-depleted serum), 100 IU/mL penicillin, 100 µg/mL streptomycin, 4 mM glutamine, and 1 mM sodium pyruvate. Cells were cultivated in a humidified incubator at 37 °C and in 5% CO_2_ atmosphere.

#### 3.2.3. AR-Transcriptional Activity Assay

ARE14 cells were seeded (40,000 cells/well) into the Nunc™ MicroWell™ 96-well optical plate (Thermo Fisher Scientific, Waltham, MA, USA) on the second day. The cultivation medium was discarded, and the cells were washed with PBS. Analyzed compounds were dissolved in medium supplemented with CSS (agonist mode) or CSS with 1 nM R1881 (antagonist mode) and added to cells, including CSS and 1 nM R1881 controls. Upon 24 h of incubation, cells were washed with PBS and lysed for 10 min in a lysis buffer (10 mM Tris pH = 7.4, 2 mM DCTA, 1% nonidet P40, 2 mM DTT) at 37 °C. Next, a reaction buffer (20 mM tricine pH = 7.8, 1.07 mM MgSO_4_^.^7H_2_O, 5 mM ATP, 9.4 mM luciferin) was added, and the luminescence was measured using a Tecan M200 Pro microplate reader (Biotek, Winooski, VT, USA).

#### 3.2.4. Cell Viability Assay

Cells were seeded into the 96-well tissue culture plates and, on the other day, compounds were added in different concentrations in duplicate for 72 h. Upon treatment, the resazurin solution (Sigma Aldrich, St. Louis, MI, USA) was added for 4 h, and then the fluorescence of resorufin was measured at 544 nm/590 nm (excitation/emission) using a Fluoroskan Ascent microplate reader (Labsystems, Budapest, Hungary). The GI_50_ value was calculated from the dose–response curves that resulted from the measurements using GraphPad Prism 5.

#### 3.2.5. Colony Formation Assay

LAPC-4 (10,000 cells per well) were seeded into 6 well plates and cultivated for 2 days. Next, the medium was removed and replaced with fresh medium containing different concentrations of the compound. Cells were cultivated with the compounds for 10 days. After that, the medium was discarded, colonies were fixed with 70% ethanol, washed with PBS, and stained with crystal violet (1% solution in 96% ethanol). Finally, wells were washed with PBS until the bottom was clear and colonies were visible and the photograph was captured.

#### 3.2.6. Immunoblotting

After the treatment, cells were washed twice with PBS, pelleted, and kept frozen in −80 °C. Cells were lysed, as usual, in ice-cold RIPA lysis buffer supplemented with protease and phosphatase inhibitors. Cells were disrupted by ultrasound sonication on ice and clarified by centrifugation at 14,000× *g* for 30 min. Protein concentration was measured and balanced within samples. Protein solutions were denatured in SDS-loading buffer, and proteins were separated by SDS-PAGE and electroblotted onto nitrocellulose membranes. Membranes were blocked in 4% BSA and incubated overnight with primary antibodies. On the next day, membranes were washed and incubated with secondary antibodies conjugated with peroxidase. Peroxidase activity was detected by SuperSignal West Pico reagents (Thermo Fisher Scientific, Waltham, MA, USA) using a CCD camera LAS-4000 (Fujifilm, Minato, Japan). Primary antibodies purchased from Merck (Darmstadt, Germany): (anti-β-actin, clone C4; anti-phosphorylated AR (S81)) and from Cell Signaling Technology (Danvers, MA, USA) (anti-AR, clone D6F11; anti-PSA/KLK3, clone D6B1; anti-Nkx3.1, clone D2Y1A; anti-rabbit secondary antibody (porcine anti-rabit immunoglobulin serum)). All antibodies were diluted in 4% BSA and 0.1% Tween 20 in TBS.

#### 3.2.7. Molecular Docking

Molecular docking of compounds **3**–**7j** and **6**–**4z** was performed into the crystal structure of CYP17A1 co-crystalised with heme and abiraterone (PDB:3RUK). The abiraterone molecule was extracted from the protein target before docking, for which the protein was set rigid. For molecular docking into the AR-LBD structure, its crystal structure with DHT was used (PDB:2PIV), and two key amino-acid residues in both extremities of the cavity (Arg752 and Thr877) were set flexible. Accuracy of the docking was assured by re-docking of abiraterone and galeterone into the protein targets and comparison with crystal structure or previously published docking poses. The 3D structures of all compounds were prepared, and their energy was minimized by molecular mechanics with Avogadro 1.90.0. Polar hydrogens were added to molecules with the AutoDock Tools program [42], and docking was performed using AutoDock Vina 1.05 [43]. Figures were generated in Pymol ver. 2.0.4 (Schrödinger, LLC, Cambridge, UK).

## 4. Conclusions

In summary, in this paper, we present the synthesis and biological studies of steroids containing an isoxazole fragment on their side chain. The presented synthetic approach allowed the preparation of regioisomeric isoxazole derivatives bearing a steroid moiety at both C-3 and C-5 of the heterocycle using common intermediates. Biological studies of the obtained compounds included an examination of their effects on 17α-hydroxylase and 17,20-lyase activity of human CYP17A1 and the ability of selected compounds to influence the downstream AR signaling.

Most of the compounds have a moderate inhibitory effect on the activity of human CYP17A1. The most promising results (predominant inhibitory effect on 17/20-lyase reaction over effect on 17α-hydroxylase activity of CYP17A1) were obtained for the compounds **41a** and **41k**. These molecules are the most perspective for further optimization. Compounds **41f**,**g**,**j** also had a predominant effect on the 17,20-lyase reaction of CYP17A1. Moreover, binding and interactions of **41a** in CYP17A1 was described using molecular docking and was found nearly identical, compared to abiraterone. Several compounds were further evaluated for their ability to affect the AR transactivation and the viability of several PCa cell lines. Within prepared compounds, three AR antagonists were found to abolish the AR transcriptional activity and the viability of AR-positive PCa cell lines in mid-micromolar concentrations. Candidate compound **24j** decreased the AR protein level and blocked its downstream signaling and significantly inhibited colony formation of LAPC-4 cells. Binding of **24j** in AR-LBD was described to be similar to galeterone. Overall, the results support the development of novel steroidal derivatives targeting CYP17A1 and AR as anticancer agents in PCa therapy.

## Data Availability

Not applicable.

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
