# Peer review of "Synthesis and Biological Evaluation of New Isoxazolyl Steroids as Anti-Prostate Cancer Agents"

_ijms, 2022, doi:10.3390/ijms232113534_

Round 1
Reviewer 1 Report
This was a really strong paper that was enjoyable to read and I think will be of interest to readers. A few minor improvements are mentioned below. One significant issue that I have found with the paper is the inclusion of the statement that supporting is included (see point 1 below). Nothing was included in submission (I clarified this with the journal). I think incorporating supporting information (NMR spectra, raw biological data) would add to the scientific rigour of this manuscript. I also don't want to suggest this for publication with the statement that supporting information is included without having seen it.
- the statement "Supplementary Materials: The following supporting information can be downloaded at: 1163 www.mdpi.com/xxx/s1." was included in the manuscript but no supporting information has been provided
- Could the name abiraterone be included near its structure in Figure 1
- Compound 7 and 21 have defined double-bond geometry. Is that confirmed, and if not please make sure that is replicated in the figure
- the formation of by-product 26 only occurred with substituents b and c. Can the authors provide some explanation/commentary as to why it occurs with these compounds but not other substituents.
- Can the authors state the methods they attempted to deprotect compound 28
- line 211 write t-butyl instead of t-but
- line 283 tert-butyl not terc-butyl
- line 1128 - compounds are miss numbered
Reviewer 2 Report
Please see my comments below.
1. Line 71: Structures of some steroidal heterocycles should be corrected as follows: Structures of some heterocyclic steroids (1-3)...
2. Scheme 1: the retrosynthesis arrow is an open arrow, please correct.
3. Scheme 1: please add the other three rings of the steroid on compounds 9 and 10.
4. The correct names for abbreviations should be mentioned once throughout the manuscript.
5. Scheme 4: compound 26b,c, the numbering should be added, because of the explanation of the structure based on NMR data.
Reviewer 3 Report
1. You presented a series of isoxazolyl steroids as potential anti-prostate cancer agents. However, the rationale to support your idea and for compound design is not sufficient. Please include more background information in your introduction section and clarify the compound design rationale in the chemistry section. In addition, please point out your main purpose and expectation for the current study at the end of the introduction section.
2. In the biology part, please add a table for all the compounds that are used for the bioassays. If you only chose selected compounds for the assays, please clarify the reasons. Meanwhile, please add the reference for the assays.
3. The SAR analysis for the CYP17A1 activity is missing. Please add it to the current manuscript.
4. For the CYP17A1 binding assay, what are the positive and negative controls? For the CYP17A1 activity assay, what's the activity of candidate compound 24j? Please clarify that. For the clonal formation assay, please include a positive control.
5. Since your compounds are steroids bearing isoxazolyl and their structures are similar to the compounds 2 and 3 in your manuscript, especially compound 3. In addition, compounds 2 and 3 have different mechanisms in downregulating AR signaling. On the contrary, galeterone bears a benzimidazole group and acts as a CYP17A1 inhibitor for prostate cancer treatment. Please briefly explain why not use compounds 2 and 3 as positive controls. In addition, please add the structure of galeterone in your current manuscript.
6. Nkx3.1 is the prostatic tumor suppressor and usually with low or loss of expression in prostate cancer cell lines. The inhibitor treatment caused the downregulation of Nkx3.1. Please briefly explain the possible reasons and add the corresponding references.
7. Many compounds are lacking the MS results for the molecular weight. Please add the MS results for these compounds. In addition, for the compounds used for bioassays, please provide the HRMS results.
8. The current manuscript has many typos, please revise the manuscript thoroughly.
Round 2
Reviewer 3 Report
I recommend accepting in present form.